# Rain event detection in commercial microwave link attenuation data using convolutional neural networks

Julius Polz[1], Christian Chwala[1,2], Maximilian Graf[1], and Harald Kunstmann[1,2]

[1]Karlsruhe Institute of Technology (KIT), Campus Alpin, Institute of Meteorology and Climate Research (IMK-IFU), Kreuzeckbahnstr. 19, 82467 Garmisch-Partenkirchen, Germany

[2]University of Augsburg, Institute of Geography, Alter Postweg 118, 86159 Augsburg, Germany

**Correspondence:** Julius Polz (julius.polz@kit.edu), Christian Chwala (christian.chwala@kit.edu)

**Abstract.** Quantitative precipitation estimation with commercial microwave links (CMLs) is a technique developed to supplement weather radar and rain gauge observations. It is exploiting the relation between the attenuation of CML signal levels and the integrated rain rate along a CML path. The opportunistic nature of this method requires a sophisticated data processing using robust methods. In this study we focus on the processing step of rain event detection in the signal level time series of the CMLs, which we treat as a binary classification problem. This processing step is particularly challenging, because even when there is no rain the signal level can show large fluctuations similar to that during rainy periods. False classifications can have a high impact on falsely estimated rainfall amounts. We analyze the performance of a convolutional neural network (CNN), which is trained to detect rainfall specific attenuation patterns in CML signal levels, using data from 3904 CMLs in Germany. The CNN consists of a feature extraction and a classification part with, in total, 20 layers of neurons and $1.4 \times 10^5$ trainable parameters. With a structure, inspired by the visual cortex of mammals, CNNs use local connections of neurons to recognize patterns independent of their location in the time-series. We test the CNNs ability to generalize to CMLs and time periods outside the training data. Our CNN is trained on four months of data from 800 randomly selected CMLs and validated on two different months of data, once for all CMLs and once for the 3104 CMLs not included in the training. No CMLs are excluded from the analysis. As a reference data set we use the gauge adjusted radar product RADOLAN-RW provided by the German meteorological service (DWD). The model predictions and the reference data are compared on an hourly basis. Model performance is compared to a state of the art reference method, which uses the rolling standard deviation of the CML signal level time series as a detection criteria. Our results show that within the analyzed period of April to September 2018, the CNN generalizes well to the validation CMLs and time periods. A receiver operating characteristic (ROC) analysis shows that the CNN is outperforming the reference method, detecting on average 76% of all rainy and 97% of all non-rainy periods. From all periods with a reference rain rate larger than 0.6 mmh$^{-1}$, more than 90% were detected. We also show that the improved event detection leads to a significant reduction of falsely estimated rainfall by up to 51%. At the same time, the quality of the correctly estimated rainfall is kept at the same level in regard to the Pearson correlation with the radar rainfall. In conclusion, we find that CNNs are a robust and promising tool to detect rainfall induced attenuation patterns in CML signal levels from a large CML data set covering entire Germany.

**Keywords:** precipitation, remote sensing, pattern recognition, deep learning, quantitative precipitation estimation

## 1   Introduction

Rainfall is the major driver of the hydrologic cycle. Accurate rainfall observations are fundamental for understanding, modeling and predicting relevant hydrological phenomena, e.g. flooding. Data from commercial microwave link (CML) networks have proven to provide valuable rainfall information. Given the high spatio-temporal variability of rainfall, they are a welcome complement to support traditional observations with rain gauges and weather radars; particularly in regions where radar is hampered by beam blockage or ground clutter. In regions with sparse rainfall observation networks, like in developing countries, CMLs might even be the only source of small scale rainfall information.

Since the work of Messer et al. (2006) and Leijnse et al. (2007) more than a decade ago, several research groups have shown the potential of CML data for hydrometeorological usage. Prominent examples are the countrywide evaluations in the Netherlands (Overeem et al., 2016b) and Germany (Graf et al., 2019), which demonstrated that CML-derived rainfall information corresponds well with gauge-adjusted radar rainfall products, except for the cold season with solid precipitation. CML-derived rainfall information was also successfully used for river runoff simulations in a pre-alpine catchment in Germany (Smiatek et al., 2017) and for pipe flow simulation in a small urban catchment in Czech Republic (Pastorek et al., 2019). A further important step was the first analysis of CML-derived rain rates in a developing country, carried out by Doumounia et al. (2014), with data from Burkina Faso.

In general, the number of CMLs available for research has increased significantly over the last years and researchers from several countries have gained access to CML attenuation data. Currently, data from 4000 CMLs over Germany is recorded continuously with a temporal resolution of one minute via a real-time data acquisition system (Chwala et al., 2016). The number of existing CMLs over Germany is 30 times higher (Bundesnetzagentur, 2017), amounting to 130.000 registered CMLs. Consequently, it is envisaged to increase the number of CMLs included in the data acquisition.

With this large number of CMLs available in Germany and with new data being retrieved continuously, there is a need for optimized and robust processing of these big data sets. Several studies address the details of the processing steps which are required for deriving rainfall information from CMLs. These steps involve, e.g. the detection of rain events in noisy raw data, the filtering of artifacts, correcting for bias due to wet antenna attenuation (WAA) and the spatial reconstruction of rainfall fields. Uijlenhoet et al. (2018) give a general overview of the required processing steps and the existing methods and Chwala and Kunstmann (2019) discuss and summarize the related current challenges.

### 1.1   On the importance of rain event detection

The first of these processing steps, called rain event detection, is the separation of rainy (wet) and non-rainy (dry) periods. A static signal level baseline to derive attenuation that can be attributed to rainfall has proven to be ineffective due to e.g. daily or annual cycles and unexpected jumps in the time series like for CML B in Fig. 1. Therefore, after the rain events are localized correctly, an event specific attenuation baseline can be determined and actual rain rates can be derived via the $k$-$R$ power law

which relates specific attenuation $k$ in dB km$^{-1}$ to rain rate $R$ in mm h$^{-1}$.

Detecting rain events is challenging, because CML signal levels can show high fluctuations, even when there is no rain, e.g. due to multi-path propagation (e.g. Chwala and Kunstmann, 2019, Fig. 6). Therefore, the main difficulty is to distinguish between noise and signal fluctuations caused by rain along the CML path. As seen in Fig. 1, the differences in noise levels can vary significantly, depending on the CML that is used. When looking at the magnitude of these fluctuations, we can see that a misclassification of wet and dry periods can easily lead to a large over- or underestimation of rainfall. These missed or falsely estimated quantities are often overlooked in scatter density comparisons of rainfall products like Figure 9 a) and b) below, which shows our own results. But when absolute amounts are compared, they represent an obvious issue with up to 30% of the total CML rainfall that can be attributed to false positives. As these misclassifications generate a bias different from the bias corrected in later processing steps like the WAA correction it is important to optimize the rain event detection as an isolated processing step first and to optimize subsequent processing steps afterwards.

## 1.2 State of the art

So far, several methods for rain event detection with CMLs have been proposed. The main difference that divides these methods into two groups, is the type of CML data that can be used to estimate rainfall. Depending on the available data acquisition, CML signal levels are either instantaneously sampled at a rate ranging from a few seconds up to 15 minutes or they are stored as 15-minute minimum and maximum values derived from a high instantaneous sampling rate in the background. In almost all cases only one of the two sampling strategies is available due to the type of data management through the network provider. The resulting rain event detection methods are highly optimized for one kind of sampling strategy and therefore in general incompatible with the other kind.

The following methods were developed for instantaneous measurements: Schleiss and Berne (2010) introduced a threshold for the rolling standard deviation (RSTD) of the attenuation time-series as a criteria to detect rain events. Despite being one of the first methods that were developed, a large part the method is still the most commonly used within the CML research community, as it was used in very recent studies from different working groups such as Kim and Kwon (2018), Graf et al. (2019) or Fencl et al. (2020). Chwala et al. (2012) introduced Fourier transformations on a rolling window of CML signal levels to detect the pattern of rain events in the frequency domain. Wang et al. (2012) used a Markov switching model, which was calibrated and validated for a single CML test site. Kaufmann and Rieckermann (2011) have shown the applicability of random forest classifiers and Gaussian factor graphs and validated their approach using 14 CMLs. Đorđević et al. (2013) used a simple Multilayer Perceptron (MLP) which was trained and validated on a single CML. Ostrometzky and Messer (2018) proposed a simple rolling mean approach to determine a dynamic baseline, also validated on a single CML. Most of these studies are based on a comparably low and sometimes pre-selected amount of CMLs ranging from one to a maximum of 50 devices, a number that is likely much larger in a possible operational setting.

As a detection scheme for 15 minute min/max sampled data with a 10 Hz background sampling rate Overeem et al. (2011) introduced the 'nearby link approach'. A period is considered wet if the increase of CML specific attenuation correlates with the attenuation pattern of nearby CMLs. They concluded that this is only applicable for dense CML networks with a high data

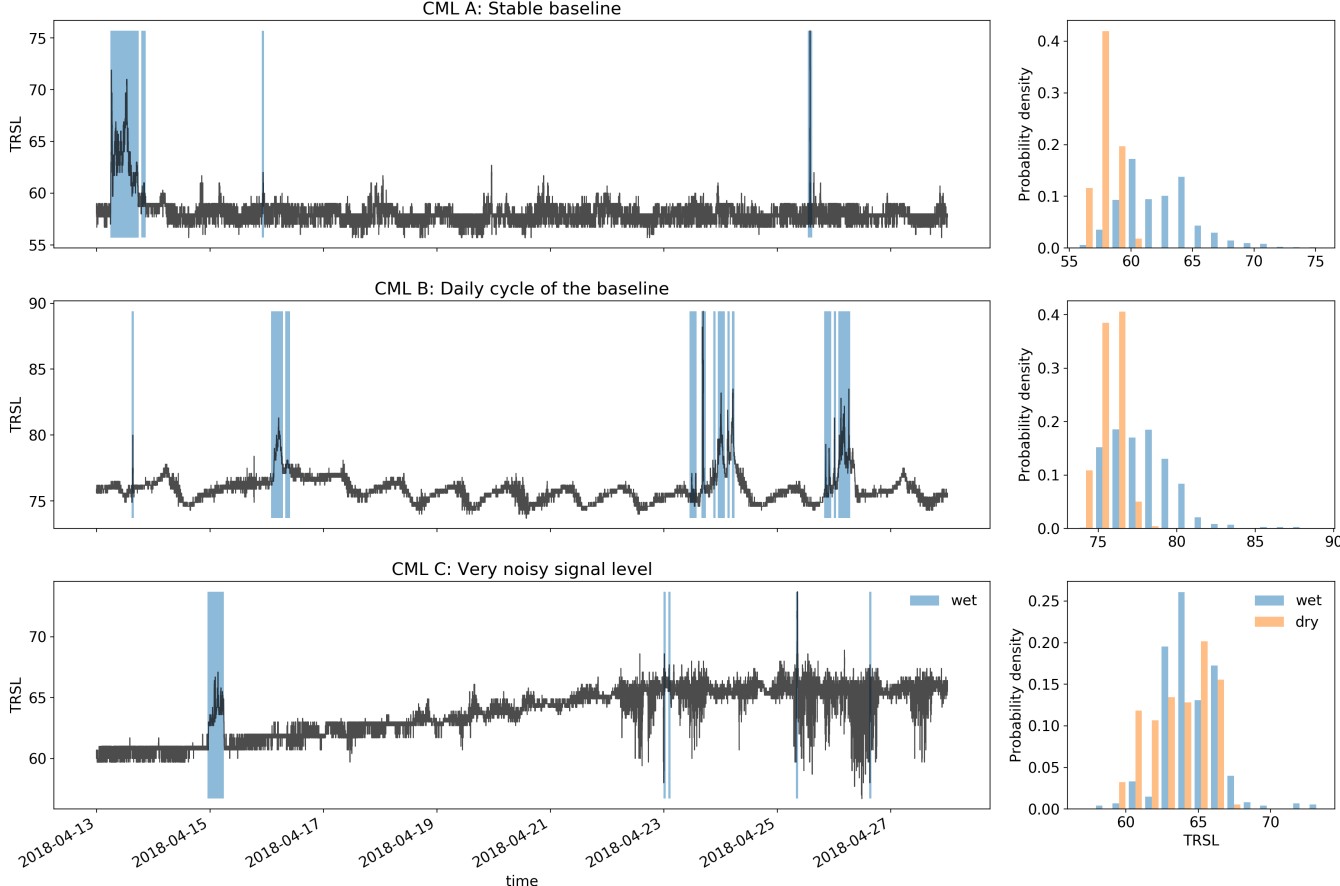

**Figure 1.** Three example signal level (TRSL) time series that illustrate the high variability in data quality when comparing different CMLs. The blue shaded periods indicate where the radar reference show rainfall along the CML paths. The challenge is to identify these periods by analysing the time series. Note that each attenuation event that is falsely classified as wet, will produce false rain rate estimates, which will lead to overestimation. The histograms show that for some CMLs the wet periods can be easily separated from the dry periods and for others the distribution of TRSL values is nearly identical for both classes. Fig. 2 below will show an example of how different detection methods deal with the challenging time series of CML C.

availability. Later, they conducted the first evaluation of a rain event detection method on data from 2044 CMLs on a country scale Overeem et al. (2016b). Very recently the same approach was used in de Vos et al. (2019), showing that this approach works better in combination with min/max sampling than with 15 minute instantaneous sampling. Habi and Messer (2018) tested the performance of Long Short-Term Memory (LSTM) networks to classify rainy periods from 15 minute min/max values of CML signal levels for 34 CMLs.

All rain event detection methods have to make a similar trade-off: A liberal detection of wet periods is more likely to recognize even small rain rates, while it will produce more false alarms during dry periods. On the other hand, a conservative detection

will accurately classify dry periods, but is more likely to miss small rain events. One can address this by two means. First, by increasing detection rates on both wet and dry periods as much as possible and therefore decreasing the impact of the trade-off. Second, by allowing the flexibility to easily adjust the model towards liberal or conservative detection, e.g. by only changing a single parameter.

In conclusion, until now, there have been few studies analyzing the performance of rain event detection methods on large data sets. Overeem et al. (2016b) tested the nearby link approach using 2044 CMLs distributed over the Netherlands with a temporal coverage of 2.5 years of data. Graf et al. (2019) extended the RSTD method and applied it to one year of data from 3904 CMLs to set a benchmark performance on the same data set used in this study. By optimizing thresholds for individual CMLs the full potential of the RSTD method for one year of data was explored, yielding good results for the warm season with liquid precipitation. While the RSTD method is simple to implement and has only two parameters (window length and threshold) to optimize, it is limited to measuring the amount of fluctuations, rather than the specific pattern. More room for optimization is expected using a data driven approach, such as machine learning techniques for pattern recognition.

## 1.3 Data driven optimization through deep learning

Deep learning is a rapidly evolving field that is becoming increasingly popular in the earth system sciences. A large field of application is remote sensing using artificial neural networks for image recognition (Zhu et al., 2017). Deep learning is also an established method in time-series classification (Fawaz et al., 2019). In both studies, convolutional neural networks (CNNs) are considered one of the leading neural network architectures for image and time-series classification. CNNs are inspired by the visual cortex of mammals and they are designed to recognize objects or patterns, regardless of their location in images or time-series (Fukushima, 1980). They are characterized by local connections of neurons, shared weights and a large number of layers of neurons, involving pooling layers (LeCun et al., 2015). CNNs with one dimensional input data (1D-CNNs) have already been used for time-series classification, e.g. for classifying environmental sounds (Piczak, 2015). This makes 1D-CNNs a promising candidate for the task of rain event detection in CML signal levels.

## 1.4 Research gap and objectives

Due to the opportunistic use of CMLs, the variety of signal fluctuations and possible occurrences of errors naturally increase in a CML data set with its size. Separating rainy from non-rainy periods is therefore a crucial step for rainfall estimation from CMLs. Although applicable on a large scale, recently applied methods still struggle with falsely estimated rainfall as can be seen in the evaluations from Graf et al. (2019) and de Vos et al. (2019). Despite the amount of proposed methods, this processing step has not yet been investigated in detail using a large and diverse CML data set, especially for data driven approaches. Given their promising results in other applications, the usage of artificial neural networks (ANNs) for rain event detection in the CML attenuation time-series on a large scale provides a promising opportunity. It has been proven that in many cases ANNs allow for high-performance, fast and robust processing of a variety of suitable data sets. What is missing is a proof that they are applicable to a large and diverse CML data set. The question is, does a high variability of frequency, length and spatial distribution of the analyzed CMLs or a high variability of rain rates and event duration for a large amount of analyzed periods

affect the performance of ANNs in this specific case or not? Additionally, the effect of rain event detection performance on the estimated rain rates has yet to be investigated.

The objective of this study is to evaluate the performance of 1D-CNNs to detect rainfall induced attenuation patterns in instantaneously measured CML signal levels and to investigate the effect of an improved temporal event localization on the CML-derived rainfall amounts. Furthermore, we test the CNNs ability to generalize to new CMLs and future time periods in order to provide a validated open source model, that can be used on other data sets. To provide the CML community with comprehensible results, we compare the CNN to the method of Schleiss and Berne (2010), which we consider state-of-the-art due to the amount of recent applications. We aim to provide a high statistical robustness of the derived performance measures by using the, to date, largest available CML data set consisting of data from 3904 CMLs distributed over entire Germany.

## 2 Methods

The following definition of rain event detection with CMLs is the basis of our methodology: Rain event detection is a binary classification problem. Given a time window $X_{t,w,i}$ of CML signal data, where $t$ is the starting time, $w$ is the window length and $i$ is the index specifying a unique CML path, we have to decide if there is attenuation caused by rain (wet) or not (dry). A time window is assigned the label 1 if it is wet or 0 if it is dry. The available information to do this classification depends on the used data acquisition and on which information is provided by the CML network operator. In the following, we describe how a CNN can be used as a binary classifier to succeed in this task.

### 2.1 Data set

We use a CML data set that has been collected in cooperation with Ericsson Germany through our custom CML data acquisition system Chwala et al. (2016). It covers 3904 CMLs across entire Germany. The CML path length ranges from 0.1 km to more than 30 km, with an average of around 7 km. CML frequencies range from 10 to 40 GHz. The acquired data consists of two sub-links per CML, transmitting their signal in opposite directions along the CML path. For each sub-link a received signal level (RSL) and a transmitted signal level (TSL) is recorded at a temporal resolution of 1 minute and a power resolution of 0.3 dB for RSL and 1.0 dB for TSL. The recorded period used in this study starts in April 2018 and ends in September 2018, to focus on the periods which are dominated by liquid precipitation, where CMLs perform better than during the cold season (Graf et al., 2019). The data is available at 97.1% of all time steps and gaps are mainly due to outages of the data acquisition system.

As reference data we use the gauge adjusted radar product RADOLAN-RW provided by the German meteorological service (DWD). It has a spatial resolution of 1x1 km, covering entire Germany on 900x900 grid cells. The temporal resolution is 60 minutes and the resolution for the rain amount is 0.1 mm (Winterrath et al., 2012). To compare to this reference, the window length $w$ is set to 60 minutes and therefore $w$ is omitted in the notation below. Along each CML $i$, the path-averaged mean

hourly rain rate $R_{t,i}$ is generated from the reference, using the weighted sum

$$R_{t,i} = \frac{\sum_k l_{k,i} r_{k,t}}{l_i},$$
(1)

where $k$ is indexing the RADOLAN grid cells intersected by the path of $i$. The rain rate of each grid cell is $r_{k,t}$. Furthermore, $l_{k,i}$ is the length of the intersect of $k$ and $i$ and $l_i$ is the total length of $i$. A time window $X_{t,i}$ is considered wet if $R_{t,i} \geq 0.1$ mm h$^{-1}$ and dry otherwise.

## 2.2 Pre-processing

Before training and testing an artificial neural network, the raw time-series data has to be pre-processed. We do this to sample time windows of a fixed size, which are normalized and labelled according to the reference.

First, the full data set, consisting of all available CMLs, is split into three subsets. One subset is used for training the CNN (TRG), one is used for validation and to optimize model hyper-parameters (VALAPR) and one is used for testing only (VALSEP). The data set TRG consists of data from 800 randomly chosen CMLs in the period from May to August 2018. VALAPR covers the remaining 3104 CMLs during April 2018 and VALSEP consists of data from all 3904 CMLs during September 2018. We used this splitting routine to avoid information leakage from the training to the validation data. There can be a high correlation of signal levels between CMLs that are situated close to each other (Overeem et al., 2011). Therefore, the measurements contained in VALAPR or VALSEP can not be taken from the same time range as for TRG. Using only 20% of all available CMLs for training allows us to analyze the CNNs generalization to the remaining CMLs in the validation data set. No CMLs were excluded from this analysis.

For each of the two sub-links of a CML, we compute a transmitted minus received signal level (TRSL). Within one TRSL time-series, randomly occurring gaps of up to five minutes of missing data are linearly interpolated to be consistent with with the preprocessing used in Graf et al. (2019). We assume that the temporal variability of rainfall is not high enough such that entire rain events can be hidden in such short gaps. The next step is to normalize the data. Normalization of training and validation data is a commonly used procedure in deep learning to enhance the model performance. We perform the normalization as a pre-processing step and outside the CNN. After testing various normalization techniques it turned out that the best performance of the CNN can be achieved by subtracting the median of all available data from the preceding 72 hours from each time step. In rare cases of larger gaps in the data acquisition, we set a lower limit for the data availability to 120 minutes.

The set of starting time-stamps of the hourly reference data set is denoted $T_{rad}$. For each CML $i$ and each starting time $t \in T_{rad}$ a sample of data $\bar{X}_{t,i}$ is composed from 60+$k$ minutes of TRSL from the two sub-links starting at $t - k$. The first $k$ minutes serve as a reference to previous behaviour of the same CML and the last 60 minutes are the period $X_{t,i}$ that has to be classified. To investigate the impact of adding this additional information, we compare multiple setups with $k$ ranging from 0 to 240 minutes. The results are given in section 3. An example TRSL over a period of two weeks is shown in Fig. 2 (a).

After interpolating short gaps, as described above, we exclude all samples with missing values from the analysis. Since we loose up to five hours of data whenever there is a gap, the interpolation routine increases the number of available samples from 75% to 94%.

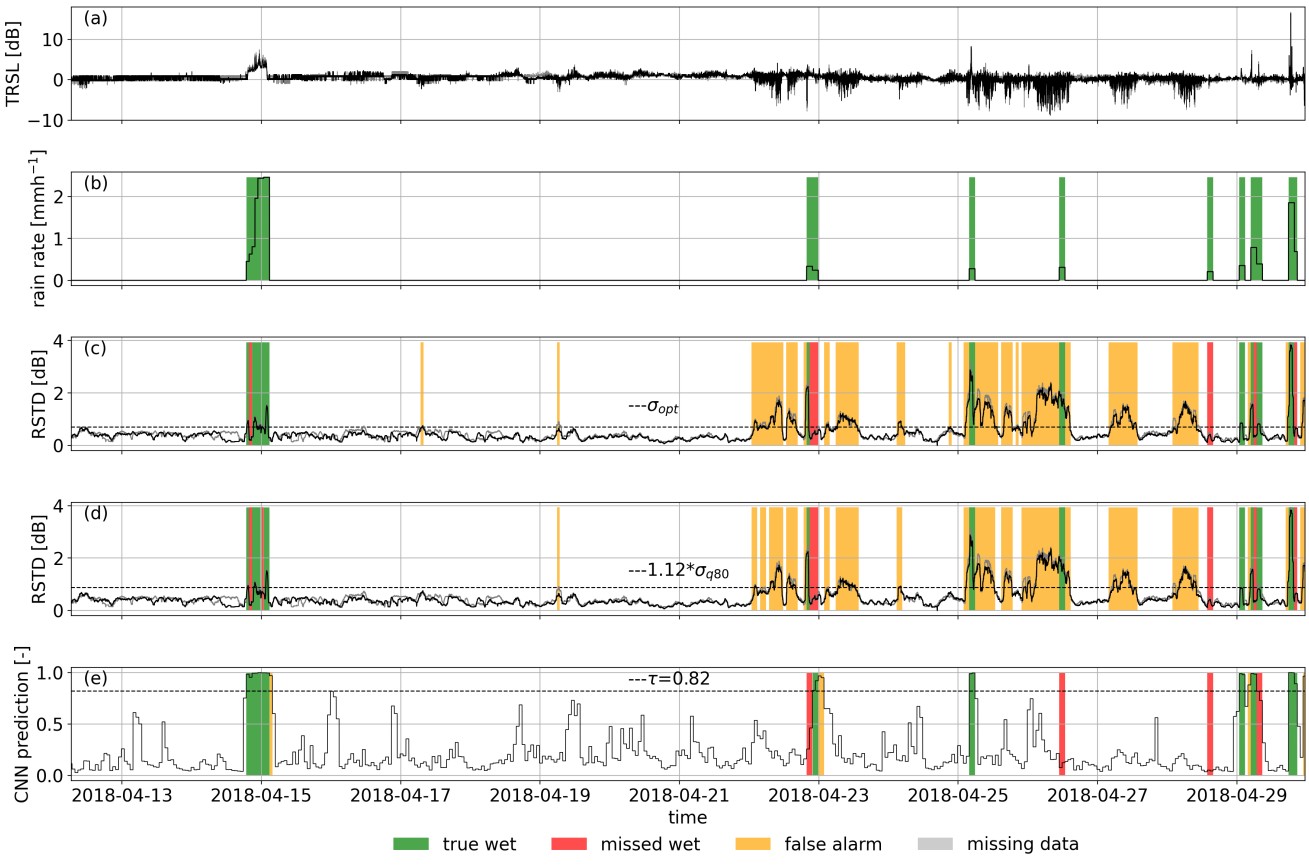

**Figure 2.** Performance of the CNN and the reference methods for the noisy example CML time-series from Fig. 1. a) shows the normalized TRSL time-series and b) is the radar reference. Predictions from the CNN (e) yield an MCC of 0.74. Predictions through $\sigma_{opt}$ (c) and $\sigma_{q80}$ (d), which are very similar in this case, both yield MCCs of 0.28. Note that the TRSL and RSTD time series of sub-link 2 are almost identical to those of sub-link 1 and are shown in light grey.

To train the CNN we have to balance the wet and dry classes in the data set (Hoens and Chawla, 2013). The under-sampling approach to achieve an equalized (50:50) class ratio is to randomly discard samples of the majority class, i.e. dry samples. This approach is chosen since we assume that dry periods mostly consist of redundant samples with only small fluctuations. Later, we check that there is no loss in performance by evaluating the unbalanced data. The initial percentage of wet samples is between 5-10%. We perform the balancing on TRG and VALAPR. The balanced version of VALAPR is denoted VALAPRB. VALAPR and VALSEP are kept as unbalanced data sets for validation. TRG already denotes the balanced data, since the original unbalanced training data set is not used in the analysis. In total, the number of samples is $2.3 \times 10^5$ for TRG, $3.9 \times 10^5$ for VALAPRB, $2.2 \times 10^6$ for VALAPR and $2.8 \times 10^6$ for VALSEP.

### 2.3 Neural Network

CNNs especially apply to time-series classification when patterns have to be recognized in longer sequences of data but the location of the occurring patterns is variable. They are therefore suitable classifiers for sensor data like the TRSL from CMLs. The expected advantage of the CNN over the reference method is that it is able to recognize the rainfall specific patterns, rather than just the amount of fluctuations. Like other neural network architectures they consist of a series of layers of neurons (Fig. 3). The first layer receives the input data and the last layer serves as an output for a prediction. The hidden layers in between are organized in two functional parts. The first part consists of a series of convolution and pooling layers and is used to extract features from the raw model input. Earlier convolution layers identify simple patterns in the data, which are used to identify more complex patterns in subsequent layers. The second part consists of fully connected layers of neurons and is used to classify the input based on the features extracted by the convolutional part.

Before a CNN can be used as a classifier, it has to be trained on data in a supervised learning process. All layers have a set of trainable parameters, so called weights, which are optimized during the training process according to a learning rule. To be able to monitor the model performance, a test data set is evaluated regularly during the training process. Training is stopped before the model starts to over-fit, i.e. the performance on the test data set either stagnates or drops, while it still rises for the training data.

#### 2.3.1 Network architecture

We use a 1D-CNN, which has the same structure as the basic 2D-CNN, with alternating convolutional and pooling layers followed by fully connected layers. The only difference is that the input data of the convolutional layers is one dimensional. The specific architecture and parameterization was optimized experimentally. To give an intuitive description of our CNN, we follow the approach provided in (LeCun et al., 2015, p. 439):

The convolutional part of the CNN consists of four blocks of two convolutional layers followed by a max pooling layer and one block of one convolutional and one average pooling layer (see Fig. 3). Convolutional layers extract feature maps by passing local patches (3x1) of input from the preceding layer through a set of filters followed by a rectified linear unit. Each filter creates a different feature map. The pooling layer then combines semantically similar features by taking the maximum (resp. average) within one local patch. This way, the dimension of the input is gradually reduced while, at the same time, the number of extracted features increases.

The fully connected part of the CNN consists of two layers with 64 neurons each and an output layer with one neuron. Its output is a prediction between zero and one, that can be interpreted as the likeliness for the input sample to be wet or dry. To avoid over-fitting to the training data two dropout layers are added, one after each fully connected layer, with a dropout ratio of 0.4 (Srivastava et al., 2014).

We implement the CNN in a Python framework using the Keras (version 2.3.1) backend for Tensorflow (version 2.1.0) (Chollet, 2015; Martín Abadi et al., 2015). For the model architecture, type, number and order of layers has to be chosen. There are several hyper-parameters that can be specified in the model setup. Each layer has a number of hyper-parameters that can be

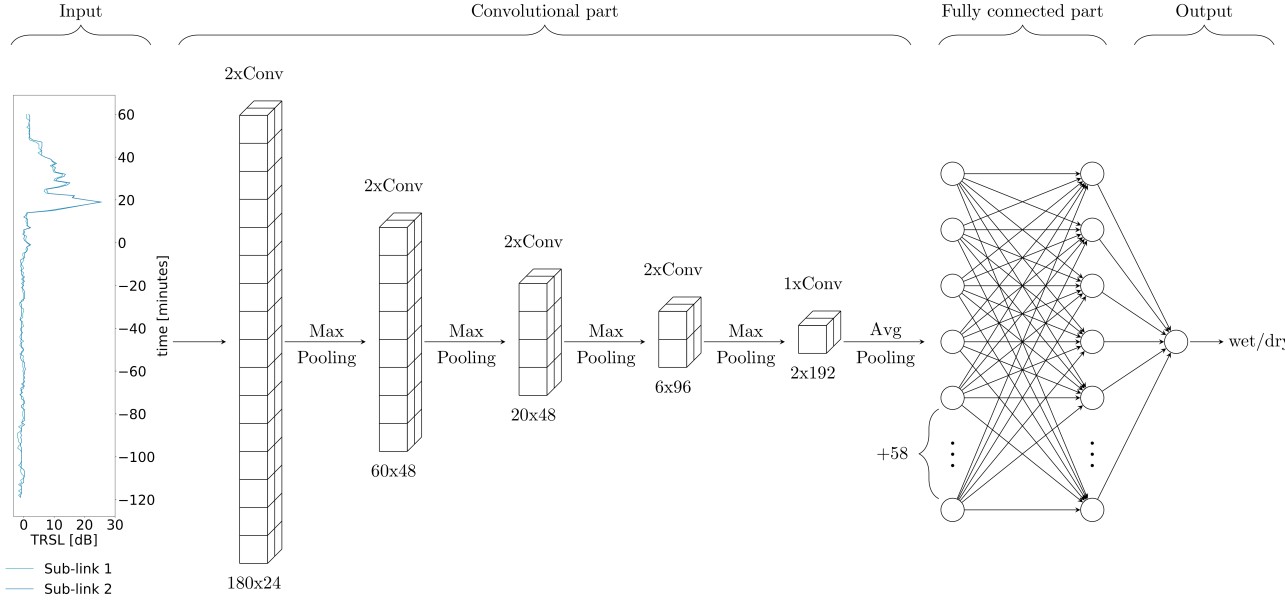

**Figure 3.** Graphical illustration of the CNNs architecture for $k = 120$. The Input shows one sample $\bar{X}_{t,i}$ of data consisting of 180 minutes of TRSL from the two sub-links of one CML. Convolutional and pooling layers reduce the input dimension from 180 to 2, while a total of 192 features are extracted. Numbers below convolutional layers are the layer output dimensions, i.e. input dimension times the number of filters. The size of the local patch in a convolutional layer is 3. Based on the extracted features, the fully connected layers predict a class, which is stored in the output layer.

adjusted, e.g. the size of the local patch or the number of filters in a convolutional layer. We optimized all hyper-parameters iteratively by evaluating the performance of several reasonable configurations on the test data set VALAPRB, and by choosing the model with the best performance metrics (see 2.4). Depending on the length of the input time-series, which varies with $k$, the number of convolutional layers is different, i.e. $k < 60$ we omit the last two convolution layers. We trained one model for
each value of $k$ and one extra model, that additionally receives the CML meta-data consisting of the length and the frequency of both channels through parallel fully connected layers and an add-layer before the fully connected part. For $k$ set to 120 minutes the final CNN consists of 20 functional layers with a total of 140,033 trainable parameters. The organization of those layers is shown in the network graph in Fig. 3. For all model versions, the detailed model and training specifications, all hyper parameters and the weights of the trained CNN can be retrieved from the code example at https://github.com/jpolz/cnn_cml_
wet-dry_example.

### 2.3.2 Training setup

CNNs are feed-forward neural networks, which are trained by a supervised learning algorithm (Goodfellow et al., 2016). Batches of samples are passed through the network and the outputs are compared to the reference labels. After each batch a

250 loss function is computed and the weights are updated according to a learning rule. Here, the learning rule is stochastic gradient descent with binary cross-entropy as a loss function and an initial learning rate of 0.008 (Bottou et al., 2018). The training data set TRG consists of 7 batches with $10^4$ samples each and the validation data set is VALAPRB. One training epoch is finished when the whole data set is used once. After each epoch the training and validation data sets are evaluated to compute the training and validation loss and the learning rate is decreased slightly.

The training is stopped if the validation loss does not equal or surpass an earlier minimal value for 50 epochs (stopping criterion). Afterwards the model which achieves the best validation Matthews correlation (see MCC below) is selected from all versions, that existed after the individual training epochs (model selection criterion). This model is then used for classification on the validation data sets.

### 2.4 Validation

Our CNN is a probabilistic classifier. The raw model output $\bar{Y}_{t,i}$ is on a continuous scale from 0 to 1 (see Fig. 5), representing the estimated likeliness that a sample $\bar{X}_{t,i}$ is wet. A threshold $\tau \in [0,1]$ is then set to decide whether a sample is wet or not, leading to the prediction rule

$$\tilde{Y}_{t,i} = \begin{cases} 1, & \text{if } \bar{Y}_{t,i} > \tau \\ 0, & \text{otherwise} \end{cases} \tag{2}$$

Classification results, in the form of true positives (TP), false positives (FP), false negatives (FN) and true negatives (TN) are

265 compared to the reference in a confusion matrix, shown in Table 1, which is the basis for computing further metrics. The normalized version of the confusion matrix consists of the occurrence rates of TP, FP, FN and TN samples, defined as

$$TPR = \frac{TP}{TP + FN}, \tag{3}$$

$$FPR = \frac{FP}{FP + TN}, \tag{4}$$

$$FNR = \frac{FN}{TP + FN}, \tag{5}$$

and

$$TNR = \frac{TN}{FP + TN}. \tag{6}$$

**Table 1.** Confusion matrix

| | | reference | |
|---|---|---|---|
| | | *wet* | *dry* |
| **prediction** | *wet* | True wet (TP): #{ detected wet| reference wet} | False wet (FP): #{ detected wet| reference dry} |
| | *dry* | Missed wet (FN): #{ detected dry| reference wet} | True dry (TN): #{ detected dry| reference dry} |

As a first metric for validation we use the accuracy score, defined as

$$ACC = \frac{TP + TN}{\text{total population}} \in [0, 1]. \tag{7}$$

It is a measure for the percentage of correct classifications being made. It is dependent on the class balance of the data set. The balance of wet and dry samples in the data set is directly related to the regional and seasonal climatology. Therefore, this metric is not climatologically independent.

The second metric we use is the Matthews correlation coefficient (MCC), also known as $\phi$-coefficient, which is a commonly used metric for binary classification (Baldi et al., 2000). It is acknowledging the possibly skewed ratio of the wet and dry periods and is high only if the classifier is performing good on both of those classes. It is defined as

$$MCC = \frac{TP \cdot TN - FP \cdot FN}{\sqrt{(TP+FP)(TP+FN)(TN+FP)(TN+FN)}} \in [-1, 1], \tag{8}$$

where an MCC of 0 represents random guessing and an MCC of 1 represents a perfect classification. A strong correlation is given at values above 0.25 (Akoglu, 2018). The advantage of the MCC is, that it is a single number which we use to optimize the threshold for the CNN.

The third metric we use is the receiver operating characteristic (ROC), defined by the pair $(FPR, TPR) \in [0,1] \times [0,1]$ (Fawcett, 2006). The domain of the ROC is called ROC space. The point (0,1) represents a perfect classifier, while the [(0,0),(1,1)] diagonal represents random guessing. The ROC is independent of the ratio of wet and dry periods and therefore a climatologically independent measure for the classifier's performance on rain event detection. Each $\tau \in [0,1]$ leads to a ROC resulting in a ROC curve $\gamma \subset [0,1] \times [0,1]$ (e.g. Fig. 4). The performance of a classifier for different values of $\tau$ is measured by the area

$$AUC = \int_0^1 \gamma d\tau \in [0, 1] \tag{9}$$

under the ROC curve. Since changing $\tau$ directly influences the prediction rule (Eq. 2), it can be adjusted causing the model to classify in a conservative (below [(0,1),(1,0)] diagonal in ROC space) or liberal (above diagonal) manner. We can therefore address the trade-off between true wet and true dry predictions as mentioned in the introduction. This way, the AUC becomes a measure of the flexibility of a classifier, i.e. the ability to show good performance with a more conservative or liberal threshold $\tau$. The main purpose of the ROC is that we use it to compare different methods, e.g. different values of $k$, independent from a fixed threshold, by considering the ROC curve and the AUC.

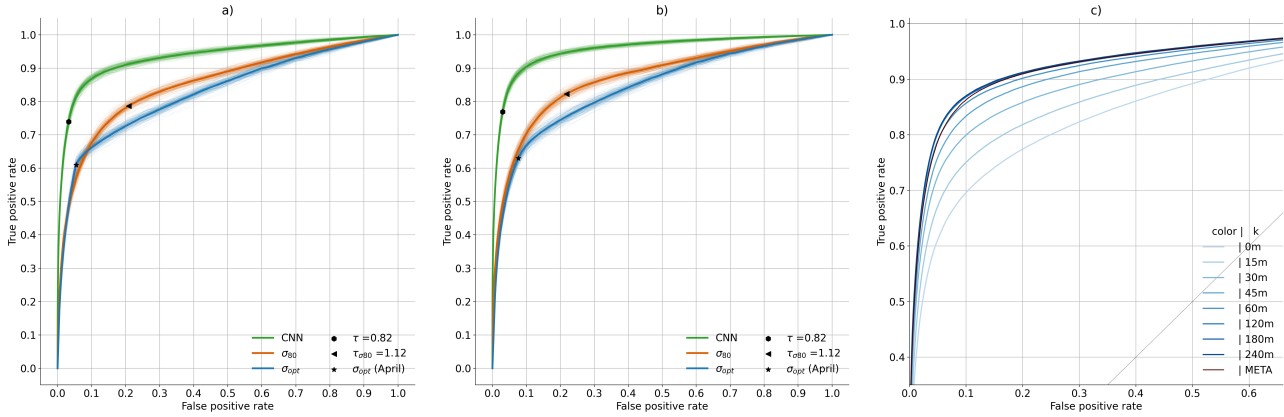

**Figure 4.** Receiver Operating Characteristic curves on VALAPR a) and VALSEP b). Fine lines are generated by 200 random selections (bootstrapping) of $1\%$ of the samples and account for the variability of the model performance during a random short period ($\sim$ eight hours) of data. The performances of the CNN for different values of $k$ and the added meta data are shown in c) and the AUC values are given in Table B1

## 2.5 Reference method

The reference method is a modification of Schleiss and Berne (2010) which is to date the most commonly used method to separate wet and dry periods as reviewed in the introduction. It is based on the following assumption: The standard deviation values of fixed-size windows of TRSL is bounded during dry periods, whereas it exceeds this boundary during wet periods and therefore allows for distinguishing the two classes. This assumption has proven to give good results on our data set, however there are known drawbacks. The method is limited to measuring the amount of signal fluctuations and there are multiple effects that can cause high signal fluctuations during dry periods, e.g. like for CML C in Fig. 1. Some of the factors are known, like multi-path propagation, but others are unknown and still need to be investigated.

The method is applied by computing a rolling standard deviation of the TRSL time-series. The normalization step is not necessary for this method. The window length is 60 minutes and the standard deviation value is written to the timestamp in the center of this window. A period $X_{t,i}$ is considered wet if at least one standard deviation value on one or both sub-links exceeds a threshold $\sigma$.

We compare two different thresholds $\sigma$, which are computed individually for each CML. The first one, denoted $\sigma_{80}$, is the 80th percentile of the 60-minute rolling standard deviation of one month for a certain CML multiplied by a scaling factor which is constant for all CMLs. In our case, the threshold is computed for VALAPR in April and VALSEP in September. The scaling factor of 1.12 is adopted from Graf et al. (2019). The second one, denoted $\sigma_{opt}$, is optimized against the reference by maximizing the MCC. We computed it for April 2018 and then reapplied it to September 2018 to test its transferability to future time periods. To derive ROC curves, we applied a scaling factor $\tau_\sigma$ to each of the standard deviation thresholds. In the following we will refer to $\sigma_{80}$ and $\sigma_{opt}$ as both the resulting detection method and the threshold.

## 2.6 Rain rate estimation

In the same way as the rolling standard deviation, the CNN can be used in a rolling window approach, classifying the timestamp
$t$ as wet or dry by using the sample with starting timestamp $t-30$ as model input. With the resulting rain event detection
information from either the CNN or the two reference methods, rain rates are estimated in several steps. We use the exact same
processing scheme as described in Graf et al. (2019), which we refer the reader to for all the technical details. This processing
includes erratic treatment of CMLs and WAA compensation to derive rain rates with a temporal resolution of one minute. For
each detected rain event a constant baseline of the TRSL is calculated from the preceding dry period. The attenuation above
this baseline level is attributed to rain but also to WAA. The WAA is compensated depending on the rain rate using a method
modified after Leijnse et al. (2008). The remaining specific attenuation $k$ is used to derive the path averaged rain rate $R$ using
the $k-R$ relation from Eq. 10. The constants $a$ and $b$ are taken from ITU (2005).

$$k = aR^b \tag{10}$$

For the CMLs used in this study this relation is close to linear, i.e. $b$ is close to one. For a comparison to RADOLAN-RW the
one minute rain rates are then aggregated by taking the hourly average.

Only from this analysis data from 45 CMLs (1.1 %) is discarded due to substantially erratic signal levels to be able to follow
the same procedure as in Graf et al. (2019). Additionally, we justify this procedure with the following observation: For the rain
event detection we want periods of erratic behavior to be included in both training and validation data, since also CMLs that are
not discarded by the erratic treatment can show periods of erratic behavior, such as CML C from Fig. 1. Each erratic training
and validation sample contributes to the final statistics as one sample and the erratic CMLs do not distort the analysis. This
is very different for the rainfall amount, since erratic links are prone to a very high overestimation of the final rain rates even
when a low amount of time periods is detected wet. Since erratic CMLs are a small fraction of the available CMLs and they
can be detected automatically, we decided to exclude their bias when analyzing the contribution of false positives to absolute
rainfall amounts. An example of such a time series can be found in Fig. A2.

## 3 Results

During training on TRG, the performance of the CNN was evaluated on VALAPRB after each epoch. The resulting graphs
of loss, ACC, TPR and TNR during the training process are shown in Fig. 6. For all three variables the performance on TRG
and VALAPRB were similar across all epochs with slightly higher performance on TRG. The threshold $\tau$ was optimized using
VALAPR, by maximizing the MCC, with resulting values of $\tau$ shown in Tab. B1. The results from that table and the ROC
curves in Fig. 4 c) show that in general the performance of the CNN is increasing with higher values of $k$, but the performance
gain was insignificant for raising the value higher than 120 minutes or adding meta data as model input. We therefore decided
to set $k = 120$ and not to use added meta data for evaluating further results and comparing them to the reference methods.
Fig. 5 shows the distribution of the CNNs predictions on VALAPRB. The threshold $\tau$ is set to 0.82. The final number of
training epochs was 248 and the model from epoch 212 was selected (see Fig. 6 (a)). On one Nvidia Titan Xp GPU the training

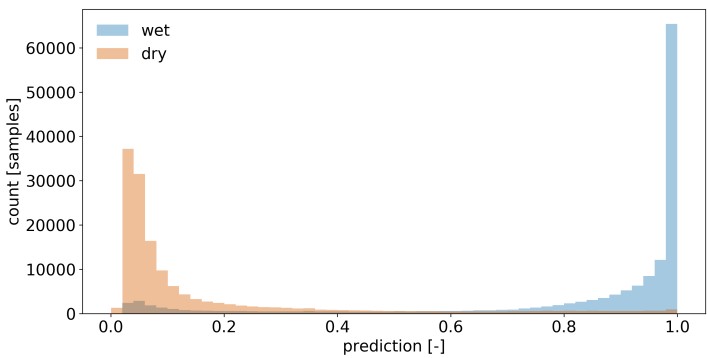

**Figure 5.** Raw CNN predictions on VALAPRB, coloured according to the reference.

**Table 2.** Performance metrics of rain event detection methods on VALAPR and VALSEP

|        | Method | TPR | TNR | ACC | MCC | AUC |
|--------|--------|------|------|------|------|------|
| *VALAPR* | CNN | 0.74 | **0.97** | **0.95** | **0.69** | **0.94** |
|        | $\sigma_{q80}$ | **0.79** | 0.79 | 0.79 | 0.38 | 0.85 |
|        | $\sigma_{opt}$ | 0.61 | 0.95 | 0.91 | 0.52 | 0.83 |
| *VALSEP* | CNN | 0.77 | **0.97** | **0.96** | **0.69** | **0.96** |
|        | $\sigma_{q80}$ | **0.82** | 0.78 | 0.78 | 0.35 | 0.87 |
|        | $\sigma_{opt}$ | 0.63 | 0.92 | 0.90 | 0.44 | 0.84 |

time was 30 minutes. Classifying 3904 samples, i.e. a one minute time-step for all CMLs, took 20ms which can be considered extremely fast allowing for a real-time application of the method. For further verification, we repeated the training multiple times with a different randomization (selection of CMLs and balancing) of TRG and VALAPRB but no significant changes in performance could be observed.

We evaluated the performance of the CNN and both reference methods using the unbalanced data sets VALAPR and VALSEP.

The complete list of the achieved performance metrics is presented in Table 2. Applying the threshold $\tau$ to the CNN predictions yielded TPRs of 0.74 (VALAPR) and 0.77 (VALSEP) and TNRs of 0.97 (VALAPR and VALSEP) (see also Fig. A1). On average, only 3% of the dry periods were falsely classified as wet and 24% of the wet periods were missed. With a scaling factor $\tau_{\sigma_{q80}}$ of 1.12, $\sigma_{q80}$ achieved a balanced TPR and TNR with a value of around 0.79 for both rates in April and September. $\sigma_{opt}$ on the other hand achieved similar TNRs than the CNN but at the cost of lower TPRs.

For both data sets, the CNN's ROC showed a higher TPR for any fixed FPR than the reference methods (see Fig. 4). As a consequence, the AUC was largest for the CNN. On VALAPR, $\sigma_{opt}$ yielded a better ROC than $\sigma_{q80}$, but only for low FPR values. On VALSEP $\sigma_{q80}$ achieved a better ROC than $\sigma_{opt}$. The ROC curves of the CNN and $\sigma_{q80}$ had a very similar convex

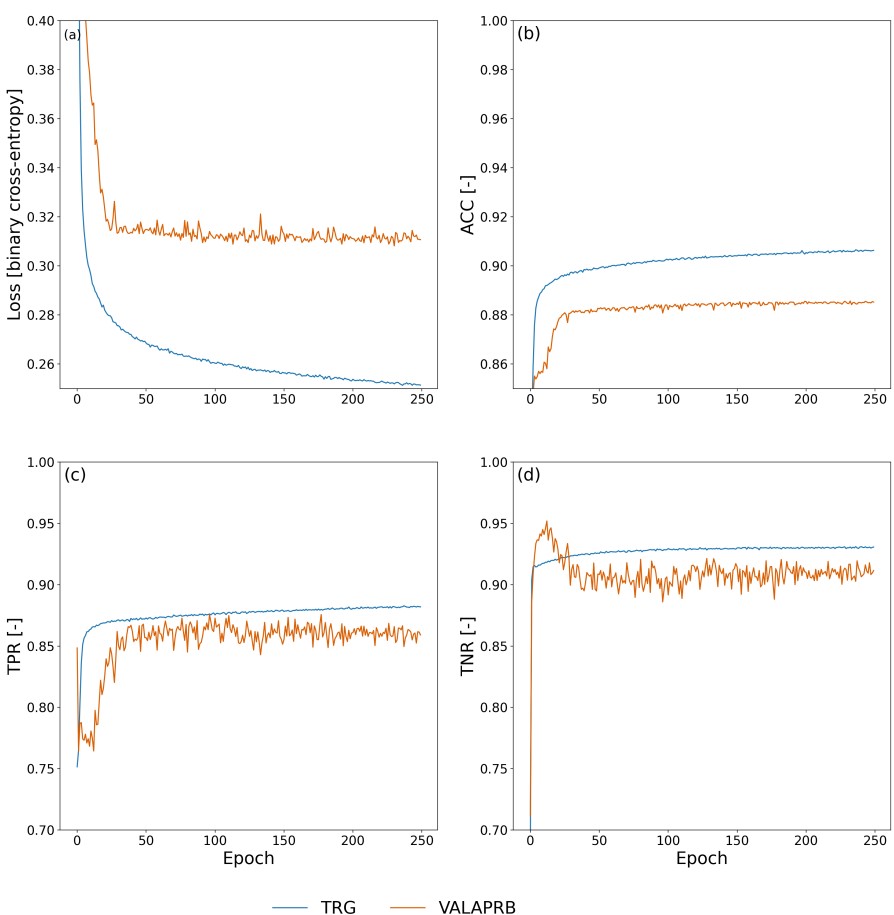

**Figure 6.** Statistics of variables that were monitored during the training process.

shape. Compared to the other two curves the ROC curve of $\sigma_{opt}$ showed a higher asymmetry. The CNN achieved the highest ACC and MCC scores with an average of 0.95 and 0.69 on both data sets. While $\sigma_{opt}$ has the second highest ACC and MCC scores, the area below the ROC curve is lowest for both data sets.

We compare the ACC on detecting samples with a specific RADOLAN-RW rain rate of $x < R_{t,i} < x + 0.1$ in Fig. 7. From all rain events where $R_{t,i} \geq 0.6\ mm$ 90.4% were correctly detected by the CNN. On the other hand around 38.9% of all rain events with $R_{t,i} < 0.6\ mm$ were missed. All three methods have a lower ACC, the lower the rain rate is. While $\sigma_{q80}$ shows an ACC for wet periods of different rain intensities, that is very similar to that of the CNN, $\sigma_{opt}$ misses more small events. On the other hand $\sigma_{q80}$ is producing more false wet classifications than the CNN or $\sigma_{opt}$.

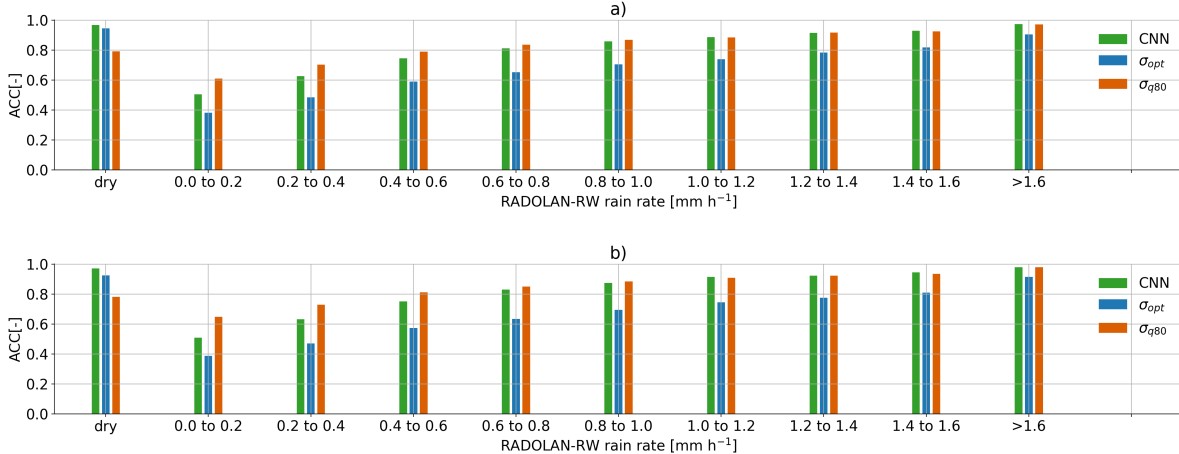

**Figure 7.** Each bar shows the ACC score on samples from a) VALAPR and b) VALSEP, grouped by the reference rain rate. An ACC of 0.5 represents random guessing.

The MCC was computed individually for each CML and each validation data set. Figure 8 shows scatter density plots comparing the individual MCC scores of the CNN and $\sigma_{opt}$. The CNN's MCC on VALAPR is higher for 95.9% of all CMLs and on VALSEP it is higher for 96.7% of all CMLs.

We focus our analysis on hourly rainfall rates from all non-erratic CMLs in September 2018. The resulting rain rates using
either the CNN or the $\sigma_{q80}$ detection scheme are shown in Fig. 9. For both methods the distribution of false positive and false negative samples is centered around 0.1 mmh$^{-1}$ and the distribution of true positives is centered around 1 mmh$^{-1}$. While the percentage of CML derived rainfall estimated during false positive events is 29.9% for $\sigma_{q80}$, it is significantly less for the CNN (see Fig. 9 d) and f). This constitutes a reduction of 51% of falsely estimated rainfall for the month of September 2018. At the same time the amount of missed rainfall is reduced by 27.5%. The amount of rainfall in the true positive category could
therefore be raised by 4.7%. The Pearson correlation for the hourly rainfall estimates between radar and CMLs is 0.83 using $\sigma_{q80}$ and 0.84 using the CNN.

## 4   Discussion

### 4.1   Performance

We evaluate the performance of the CNN to detect rain events by two means. First, we compare it to the performance of
a reference method. Second, we estimate if the model is performing in a near optimal state or if we expect that a higher performance could be achieved. The comparison to the results of previous studies, e.g. Overeem et al. (2016a), is difficult since the overall performance is depending on the distribution of the intensity of rain events (see Fig. 7) and since there is a large

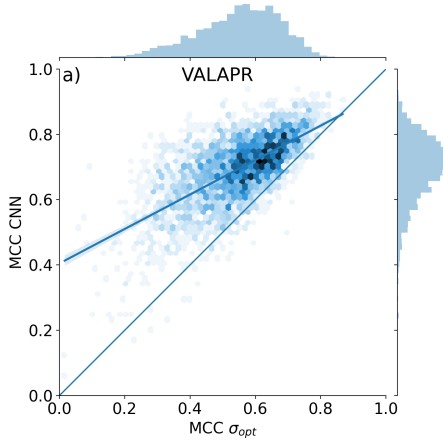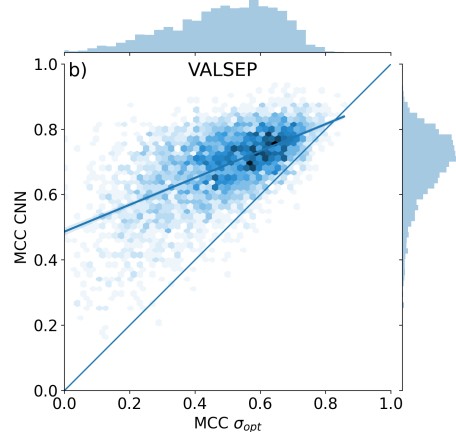

**Figure 8.** Scatter density plots of the MCC achieved by the CNN and $\sigma_{opt}$ on data from individual CMLs. Both methods are MCC optimized for the unbalanced data from VALAPR, while the CNN keeps the optimized performance in September, the performance of $\sigma_{opt}$ drops.

variability of performance between the CMLs (see Fig. 8).

Since the results on both validation data sets are very similar (see Table 2) we further focus on VALSEP, which was not used to
optimize the model hyper-parameters. With an ACC of 0.95 and an MCC of 0.69 the correlation between the CNN predictions and the reference data set RADOLAN-RW can be considered as very high. A TPR of 0.74 might not appear very good at first sight, but considering that the detection accuracy for samples with a rain rate of smaller than 0.6 mmh$^{-1}$ is only 0.61, we actually achieve an accuracy of over 0.9 for all rain rates higher than 0.6 mmh$^{-1}$.

The CNN and the reference method $\sigma_{opt}$ have a similar ACC value. At the same time the CNN's MCC is higher, despite the fact
that $\sigma_{opt}$ is MCC optimized for each CML. The high ACC of $\sigma_{opt}$ is due to the high TNR and the fact that 95% of all samples are negative (dry). At a similar ACC and TNR we could increase the TPR, or rain event detection rate, by 0.13. This constitutes a major improvement by the CNN. As shown in Fig. 8 the improvement is higher for CMLs with lower MCC, making the whole CML data set more balanced in performance and therefore more trustworthy for quantitative precipitation estimation. The CNNs distribution of MCC values of individual CMLs is the same in April and September, while performance drops for
$\sigma_{opt}$. The CNN's improvement in ACC and MCC over $\sigma_{q80}$ was even higher with 0.17 and 0.32. While the TPR of $\sigma_{q80}$ is slightly higher than the TPR of the CNN, the TNR is much lower for $\sigma_{q80}$. Thus the CNN shows substantial improvement in correctly classifying dry periods.

While the RSTD method can be set up to either have a high TPR ($\sigma_{q80}$) or a high TNR ($\sigma_{opt}$), the ROC curves show that

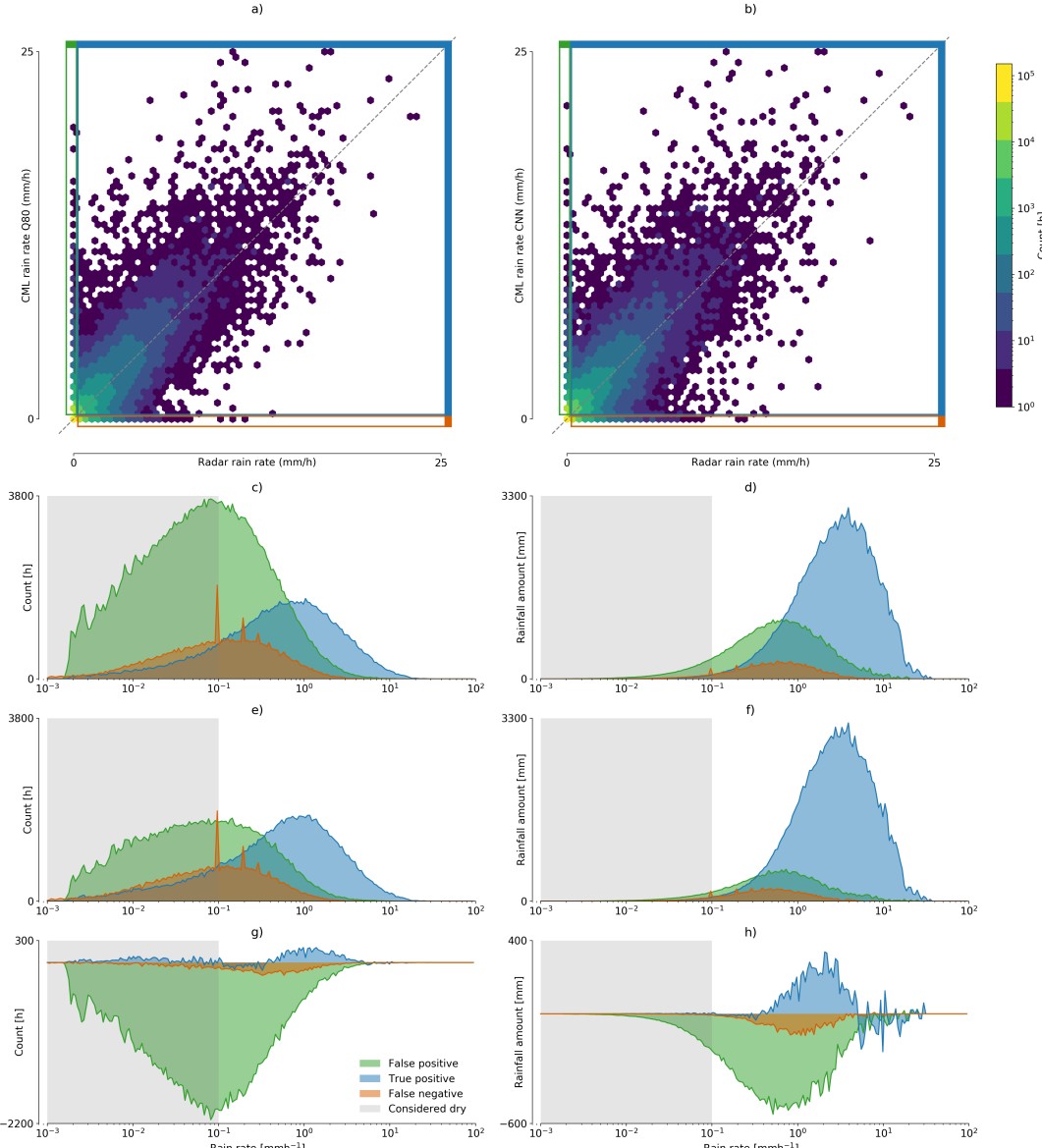

**Figure 9.** Scatter density comparison between hourly CML and radar rain rate estimates derived from a) $\sigma_{q80}$ and b) the CNN. On the left hand side the amount of FP, TP and FN hours with a specific rain rate are compared for c) $\sigma_{q80}$, e) the CNN and g) their difference). On the right hand side the amount of rainfall these hours contribute are shown for d) $\sigma_{q80}$, f) the CNN and h) their difference. The rain rates for false positives and true positives are estimated by the CML, while the rain rates for false negatives are taken from the reference.

CNN achieves both rates at the same time. Thus, the CNN shows a better overall performance than the reference methods and
therefore improves on the trade-off as mentioned above. This observation is illustrated by the example in Fig. 2, which shows
a very noisy CML time-series that produces a high amount of false positives for the reference method, while the CNN does not

attribute these fluctuations to rainfall.

All three methods have limitations to detect events with rain rates smaller than 0.3mm. This is likely due to the detection limit of CMLs in our data set which is in the same range. The detection limit depends on frequency, length and signal quantization of a CML. For example, at a frequency of <20 GHz and at a length of <10 km a path averaged rain rate of 1 mm h$^{-1}$ creates a maximum of 1 dB of attenuation (Chwala and Kunstmann, 2019, Fig. 7). In some cases the quantization (0.3dB for RSL and 1dB for TSL) might therefore not allow for a detectable signal.

Differences in the performance on VALAPR and VALSEP can be traced back to a different distribution of occurring rain rates. While in April 35.5% of all events are in the critical range from 0.1mm to 0.3mm, there are only 32% in September. In both data sets the performance on higher rain rates ($> 1.6$ mm) and dry periods is almost identical. Therefore the loss of performance in April is due to the slightly worse performance of the CNN on smaller rain rates which occur more often in VALAPR than in VALSEP.

It should not be expected that the rain events detected through CMLs and the events detected by the radar coincide completely. Both methods produce artifacts that are mistaken as rainfall, or they miss events due to their detection limits. From all false classifications that the CNN makes on VALSEP there are 50% with a raw model output between 0.2 and 0.8. Here the CNN does not give a certain prediction. This is due to very similar signal patterns in noisy dry periods and small rain rates. The other 50% of those samples are, according to the CNN, very likely to belong to the falsely predicted class. Despite this being an issue for many CMLs about 10% have a ROC of ($> 0.97, < 0.1$) and correlate very well with the RADOLAN reference. Therefore, we expect that less errors could be made when training with a perfect reference data set, but there would still be errors due to artifacts or insensitivity in CML measurements.

Despite those errors, which occur mostly for small rain rates, the correlation of wet and dry periods between RADOLAN-RW and our CML data set is very high. The performance boost in rain event detection gained through the CNN is very promising for future applications in quantitative precipitation estimation with CMLs.

## 4.2 Robustness

The CNNs ability to generalize to previously unknown CMLs is very high. As seen in the training results the learning curves for both training and validation show a similar dynamic (see Fig. 6). As expected the training data showed better performance, but the validation was close at all epochs.

Only 20% of all available CMLs were used for training. The remaining 80% were only used to prevent the model from overfitting to the training data, to choose the model architecture and to optimize the single parameter $\tau$. Thus no information about the validation data was given directly to the model. The resulting model architecture and hyper-parameters are not specific enough to store this information. The high performance in ACC, MCC and ROC on data set VALAPR, together with the learning curves in Fig. 6), therefore prove that the CNN was able to recognize the attenuation pattern in the signal levels of a large number of previously unknown CMLs.

The stability of the CNNs performance for future time periods is analyzed using the results on VALSEP. While the training was done with TRG including the period of May to August 2018, the performance in September was similar. Compared to the

results on VALAPR the CNN shows even higher performance on VALSEP, which can be explained by the lower percentage of samples with small rain rates in September, which are challenging to classify (see Fig. 7 a)). When we compare the CNNs accuracy per rain rate between VALAPR and VALSEP, we see that there are no major differences in the individual scores. Therefore the method can be considered as very stable throughout the analyzed time period, while differences in overall performance mostly stem from different distributions of the occurring rain rates. The reference method $\sigma_{opt}$, which was optimized in April, loses performance in September, where it is outperformed by the adaptive method $\sigma_{q80}$. The bootstrapping in Fig. 4 shows that all three methods perform almost equally well on small random subsets of the validation data. The CNN shows the lowest variability.

As a measure for the flexibility of a classifier we adopted the ROC analysis in section 2.4. A model is called flexible if it has a high area below its ROC curve and if the curve is axis-symmetric with respect to the [(0,1),(1,0)] diagonal of the ROC space. As observed both the CNN and $\sigma_{q80}$ show a symmetrical ROC curve. Therefore they perform almost equally well with a liberal or conservative threshold with a slight tendency to the conservative side. On the other hand $\sigma_{opt}$ shows a skewed performance, with a strong tendency to the conservative side. The area AUC below the ROC curve was highest for the CNN, making it the most flexible classifier. We can adjust $\tau$ for a ROC of either $(0.03, 0.7)$ or $(0.3, 0.94)$ and a smooth, concave transition in between (see Fig. 4).

We conclude that within the analyzed period the CNN shows a temporally stable performance, with a good generalization to previously unknown CMLs. The $\sigma_{opt}$ method performs well only if it is re-calibrated for different months and to individual CMLs, while $\sigma_{q80}$ is by definition an adaptive method. Even with re-calibration or adaption, the reference methods are outperformed by the CNN.

## 4.3 Impact of the detection scheme on the derived rainfall amounts

The difference between the scatter density plots in Fig. 9 a) and b) seems to be quite low at first sight. What this representation of the data is not stressing enough is the amount of rainfall generated by false positives. But they are an issue that is clearly visible from Fig. 9 c)-h). Considering that the amount of rainfall estimated during time periods falsely classified as wet can be reduced by 51.0% and that the amount of rainfall from missed events can be reduced by 27.4%, the CNN shows a major improvement over the reference method. The 4.1% of additional rainfall in the correctly classified wet periods stem from time periods that were originally harder to classify, i.e. from small rain events, and it should be expected, that the correlation between CML and radar rainfall drops. Instead, the Pearson correlation coefficient increased slightly showing that the quality of the estimated hourly rainfall could be improved. We omitted the same analysis for a comparison of the CNN and $\sigma_{opt}$ for which, based on the ROC values in Fig 4, we anticipate a similar result, but with a higher pronunciation of missed rain events instead of the strong impact of false positives.

Overall, we could observe that the improvement in rain event detection has a considerable effect on the amount of over- or under estimation through falsely detected or missed rain events. The improvement on the trade-off between false positives and false negatives directly translates to the impact of their respective rainfall amounts. This is shown by the false positive and false

negative distributions in Fig. 9 c)-f) which are centered around the same value, but are different in their amount depending on
the used detection method.

## 5 Conclusions

In this study, we explore the performance and robustness of 1D-CNNs for rain event detection in CML attenuation time-series using a large and diverse data set, acquired from 3904 CMLs distributed over entire Germany. We prove that, compared to a reference method, we can minimize the trade-off between false wet and missed wet predictions. While the reference method needs to be adjusted for different months of the analyzed period to provide optimal results, the trained CNN generalizes very well to CMLs and time periods not included in the training data. On average, 76% of all wet and 97% of all dry periods were detected by the CNN. For rain rates higher than 0.6 mmh$^{-1}$ more than 90% were correctly detected. This underlines the strong agreement between rain events that can be detected in the CML time-series and rain events in the RADOLAN-RW data set.

In future work, we plan to investigate the potential of using reference data with higher temporal resolution to improve the temporal localization of the rain events. Data with higher temporal resolution will, however, magnify the uncertainties that arise due to the different spatial and temporal coverage of the different rainfall observation techniques. In order to address these uncertainties, it will be important to further explore the relationship between weather radar and CML derived rainfall products. In the study presented here, we focused on the optimization of rain event detection as an isolated processing step, which provides the basis for a successful rain rate estimation. All subsequent processing steps, including WAA correction, $k$-$R$ relation and spatial interpolation, have an effect on the CML derived rain rate, that can also lead to over or under-estimation. While 29.9% of the estimated rainfall through the reference method can be attributed to false positive classifications, the CNN reduces this amount by up to 51% and, at the same time, improves on true positive and false negatives. We anticipate, that this improvement will lead to new insights into other effects that may disturb the quality of this opportunistic sensing approach.

Our study shows that using data driven methods like CNNs in combination with the good coverage of the highly developed weather radar network in Germany can lead to robust CML data processing. We anticipate that this robustness enhances the chance that we can transfer processing methods to data from other CML networks, particularly in developing countries like Burkina Faso, where rainfall information is still scarce despite its high importance to the local population (Gosset et al., 2016).

*Code and data availability.* Interactive code to build the CNN and an example evaluation using the trained CNN are available at https://github.com/jpolz/cnn_cml_wet-dry_example. CML data was provided by Ericsson Germany and is not publicly available in its full extent. RADOLAN-RW is publicly available through the Climate Data Center of the German Weather Service (DWD) https://opendata.dwd.de/climate_environment/CDC/grids_germany/hourly/radolan/. We include a small example data set with modified CML locations, the trained model weights and the pre-processed RADOLAN-RW reference data together with the interactive code at https://github.com/jpolz/cnn_cml_wet-dry_example.

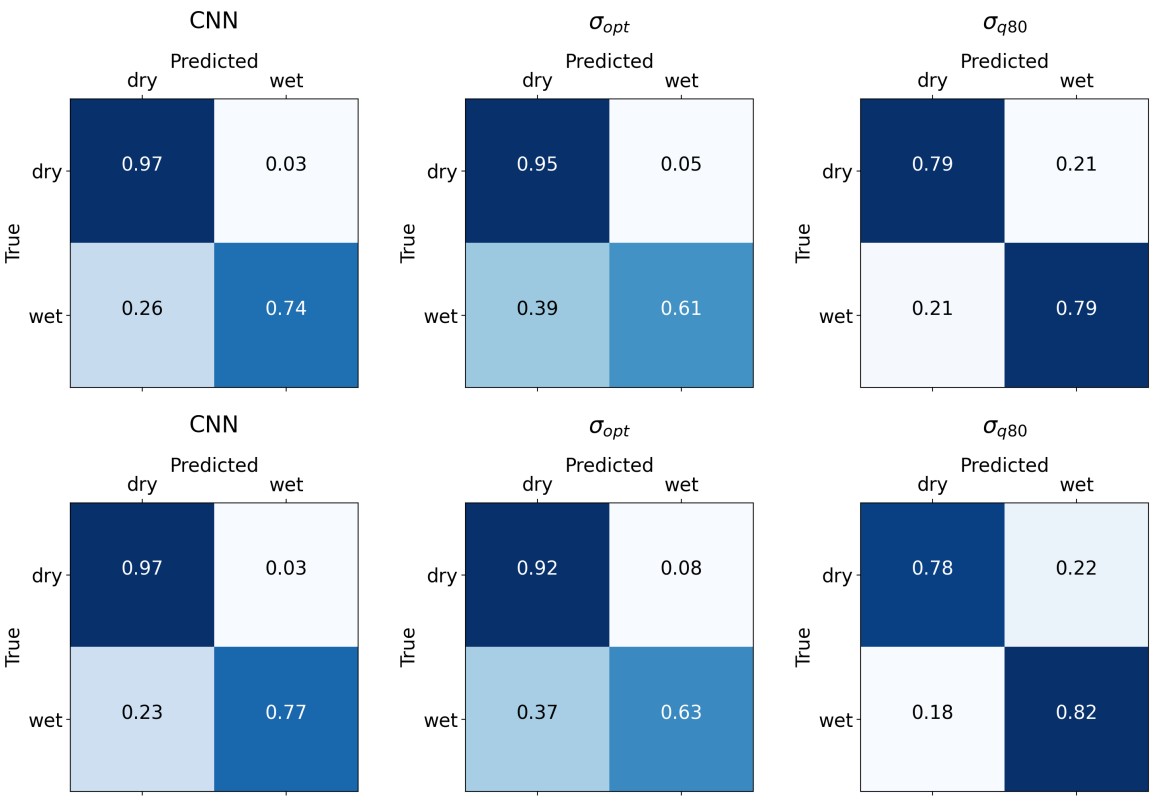

**Figure A1.** Normalized confusion matrices of VALAPR (top) and VALSEP (bottom).

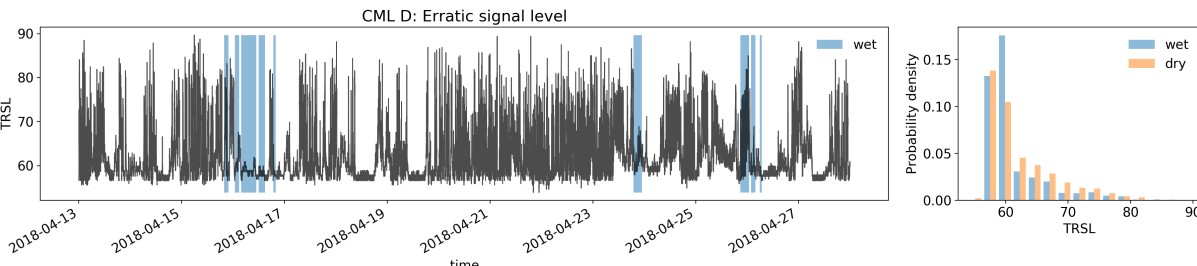

**Figure A2.** Time series of a CML that is considered as erratic and is removed by the simple filter for erratic CML data introduced in Graf et al. (2019). There are no time periods, where a reasonable rainfall estimation would be possible.

**Table B1.** Number of training epochs, MCC optimized threshold and resulting metrics for different values of $k$, evaluated on VALAPR.

| Method | $k$ | Training epochs | Threshold $\tau$ | TPR | TNR | ACC | MCC | AUC |
|---|---|---|---|---|---|---|---|---|
| CNN | 0 | 269 | 0.77 | 0.53 | 0.97 | 0.93 | 0.55 | 0.86 |
| | 15 | 158 | 0.78 | 0.59 | 0.97 | 0.94 | 0.60 | 0.88 |
| | 30 | 274 | 0.79 | 0.64 | 0.97 | 0.94 | 0.64 | 0.91 |
| | 45 | 271 | 0.79 | 0.67 | 0.97 | 0.94 | 0.66 | 0.92 |
| | 60 | 128 | 0.84 | 0.71 | 0.97 | 0.95 | 0.68 | 0.93 |
| | 120 | 212 | 0.85 | 0.72 | 0.97 | 0.95 | 0.69 | 0.94 |
| | 180 | 211 | 0.86 | 0.72 | 0.97 | 0.95 | 0.69 | 0.94 |
| | 240 | 170 | 0.84 | 0.73 | 0.97 | 0.95 | 0.69 | 0.94 |
| CNN+Meta | 180 | 321 | 0.79 | 0.70 | 0.97 | 0.95 | 0.68 | 0.93 |
| $\sigma_{q80}$ | - | - | - | 0.79 | 0.79 | 0.79 | 0.38 | 0.85 |
| $\sigma_{opt}$ | - | - | - | 0.61 | 0.95 | 0.91 | 0.51 | 0.83 |

# Appendix A: Additional Figures

# Appendix B: Additional Tables

*Author contributions.* JP, CC and HK designed the study layout and JP carried it out with contributions of CC and MG. Data was provided by CC and MG. Code was developed by JP with contributions of CC. JP prepared the manuscript with contributions from all co-authors.

*Competing interests.* The authors declare that they have no conflict of interest.

*Acknowledgements.* We thank Ericsson, especially Reinhard Gerigk, Michael Wahl and Declan Forde for their support and cooperation in the acquisition of the CML data. This work was funded by the German research foundation within the RealPEP research group. Furthermore, we like to thank the German Research Foundation for funding the project IMAP, the Helmholtz Association of German Research Centres for funding the project Digital Earth and the Bundesministerium für Bildung und Forschung for funding the project HoWa-innovativ. Special thanks are given to Bumsuk Seo for his valuable advice and for providing the Titan Xp GPU used for this research, which was donated by the NVIDIA Corporation.

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
