# Peer review of "Rain event detection in commercial microwave link attenuation data using convolutional neural networks"

_Atmospheric Measurement Techniques, 2019_

## Referee Comment (RC1) · Anonymous Referee #1 · 23 Jan 2020

In general, this paper demonstrates the use of a 1D convolution neural network for the task of Wet-Dry classification using commercial microwave links (CMLs). The scientific significance of this paper is in presenting the potential of the suggested method for the specific application: the use of 1D CNN for wet-dry classification with commercial microwave links. But, without any theoretical justification for the use of 1D CNN, it must be compared empirically with other algorithms/methods. The results are shown in this work only compare 1D CNN with a model-driven method [1], however, the suggested method must be compared with another data-driven algorithm, previously suggested (and cited by the author) - the use of LSTM for wet-dry classification [2]. This comparison is important also since the LSTM can capture long sequence, while the 1D

CNN only see a fix window size of the attenuation time series. Additionally, the authors didn't use the CML's parameters (e.g. link length, frequency, and polarization ) as an additional input to the neural network, which may make this method more sensitive to differences in those parameters.

Specific comments required for the paper to be acceptable for publication:

1. Comparing the results of the LSTM and CNN on the same data set is essential.

2. Study the effect of different window size on the performance of the proposed method.

3. Study the effect of different CML's parameters (e.g. link length, frequency, and polarization).

[1] Schleiss, M. and Berne, A.: Identification of Dry and Rainy Periods Using Telecommunication Microwave Links, IEEE Geoscience and Remote Sensing Letters, 7, 611–615, https://doi.org/10.1109/LGRS.2010.2043052, 2010.

[2] Habi, H. V. and Messer, H.: Wet-Dry Classification Using LSTM and Commercial Microwave Links, in 2018 IEEE 10th Sensor Array and Multichannel Signal Processing Workshop (SAM), pp. 149–153, https://doi.org/10.1109/SAM.2018.8448679, 2018

---

## Referee Comment (RC2) · Andreas Scheidegger (Referee) · 3 Feb 2020

* General

The manuscript describes the application of a one-dimensional convolutional neuronal network (CNN) to classify wet and dry periods based on microwave link attenuation data. The CNN is compared against a very simple classification scheme that is only based on the standard deviation of the signal. Not surprisingly, the CNN performed better.

The manuscript is well written and the underlying work seems solid. Still, in my opin-

[Figure]

AMTD

ion, this paper lacks ambition and innovation to deserve a publication in AMT. As the authors mention, various ANN's and other machine learning techniques have previously been applied in different settings on MWL data. Also the whet/dry classification problem does not appear particulary challenging from a machine learning perspective. Furthermore, for time-series data recurrent neuronal network architectures (e.g. LSTM) seem a more obvious choice (which could be combined with convolution layers if needed).

A more interesting question would be to investigate if we can train a ANN to predict the rainfall intensities directly, and so avoid all submodels for baseline estimation, wet antenna correction, and so on. Such a model could also make use of additional information, like MWL properties, frequency, maybe temperature, ...

A good transferability of the trained model to a region with different climate is key for an application where no reference data are available (such as in the mentioned Burkina Faso). This could have been partly emulated by training the model in one region and then validating it in a region with different climate. Or by training the model in winter and validating it in summer.

I'm sure the current work offers the authors a solid foundation for more ambitious investigations.

* Specific points

L 90: "...it has to be proven that artificial neural networks allow for high-performance, fast and robust processing of large data sets..." - I think this is already proven by countless other application.

L 145: Besides the attenuation data for the hour to classify, the Network was also feed with the two hours of "old" data. Did this improve the classification? If yes, it would indicate that attenuation data have some kin of memory effect (antenna wetting?).

L185: Where did you add the dropout layers? How many?

[Figure]

L216: Are TP, FP, FN, and TN defined?

L230: What is the advantage of the MCC compared to the ROC?
* * *

---

## Author Comment (AC1) · 13 Feb 2020

**Authors response to general comments by Andreas Scheidegger**

Dear Andreas Scheidegger,
Thank you for reviewing our manuscript and for providing your criticism which made us reflect our analysis and in particular how we justify its relevance. Here, we want to briefly respond to the individual general points of your review. Please note that our suggestion for specific changes, adjustments and extension for the revised manuscript will be provided in our final response, after the open discussion period.

[Figure]

**1. Importance and relevance of the rain event detection in TRSL time series data:**

**A:** Based on our experience: We acquire data of 4000 CMLs with one minute resolution in real-time every minute and we work towards providing real-time rainfall estimation from that data. From the experience that we gathered doing this over the last years, we can state that detecting rain events, or more specifically, separating TRSL fluctuations during dry times from those during rainy times, is key to produce reliable rainfall estimates. If CML rainfall estimates shall be operationally applied, missed events (false negative) and false rain event (false positive) must be minimized as good as possible.

**B:** Based on the experience of the CML rainfall community: We have presented the work that is summarized in the current manuscript in June 2019 at the *symposium on the hydrometeorological usage of data from commercial microwave link networks* which was attended by many colleagues that actively work with CML data for rainfall estimation. As far as we can recall now, the dedicated analysis of rain event detection and our approach with CNNs was appreciated by our colleagues, except for the mentioning that LSTMs are maybe better suited for time series classification (for our response to that, see point 3 in this response letter). To our knowledge only two other research groups worldwide currently acquire a CML data set with a temporal resolution and absolute size similar to that of our CML data set. At SMHI data from several hundred CMLs is acquired at subminute-scale and processed to rainfall estimates (*link to project website*). At CVUT (Czech Technical University in Prague) a similar project called *tel4rain* is ongoing. Reflecting on your review, we have contacted both groups to discuss if research for the processing step of rain event detection in TRSL time series is still required and relevant. Both groups confirmed that they consider improvements in rain event detection in TRSL time

series still very relevant, in particular when working towards operational usage of the CML derived rainfall estimates. We very much appreciate the detailed feedback on this question from Martin Fencl (martin.fencl@cvut.cz) from CVUT, who agreed to appear here with his contact details. Please note that we only reached out to discuss this specific issue and not the scientific details of our approach, using CNNs.

Again, we thank you for drawing our attention to the fact that we failed to justify the relevance of our results to the wider hydrometeorological community. This paper exists to validate a method that is about to be used within our community and to set a standard for benchmarking CML processing, which is very likely to keep the localization of rain events as an isolated processing step (see 4.). We will do our best to revise the manuscript accordingly using further argumentation as written below. We also want to clarify that this is not an effort to minimize work that we have to invest in revising the paper, but an effort to justify the topic and the relevance of the paper. We are happy to receive further constructive suggestions (like in 5.) on how to improve our manuscript.

**2. Ambition and innovation:**

First of all, we have to apologize for a mistake, which may be important in this discussion. In our review of previously used methods for the task of wet/dry classification, we wrote that Kim and Kwon 2018 [1] made use of LSTM networks. It turned out that this is not the case and that they also used a rolling standard deviation as their main criteria to separate wet and dry periods, similar to the method we compare to. This reduces the previous attempts to use deep learning for the task with no peer reviewed studies using more than one CML, that we know of. We believe that our work is novel in the following aspects:

**A:** This is the first time a large CML data set is used in combination with a data driven method. Previous attempts ranged from using a single CML up to the 34 CMLs used by Habi and Messer 2018 [2]. It is important to say that the CML data used by Habi and Messer is different from our data set since they use 15 minute min/max values for attenuation and information about different types of errors while we use instantaneous measurements at a one minute resolution. Therefore, not counting their approach to do the same data processing as we do, this paper is the first of its kind that uses more than one sensor. Many other groups also use instantaneously measured CML signal levels, but they might only have access to a handful of CMLs. Therefore, we believe that it is important for the scientific community to be able to use a model trained on our large network. With the initial submission we already made our trained model available via zenodo and github.

**B:** Though indeed very simple, methods like the rolling standard deviation, rolling median or correlation to neighbouring CMLs are still state of the art due to their known stability and easy applicability, although being less performant. A recent example is the analysis of country-wide CML data in the Netherlands by de Vos et al 2019 [3]. They use a rain event detection based on correlation between neighbouring CMLs. As their Fig. 4 shows, their approach works in general, but the amount of points along the y-axis clearly shows that false positives (falsely detected rain events) are an issue. Our work is the first evaluation of a data driven classifier, where the robustness of the method is a central point of the evaluation and proven on a large and diverse data set, facilitating the application for other researchers and operational users.

In conclusion, we understand that our review of previously used methods is lacking detail and does not manage to convey the still persisting challenges in rain event detection. We will better describe the state of the art and improve our explanations on how we advance it with our work.

**3. CNN vs. LSTM**

It is true that recurrent neural networks (RNNs) are a dedicated tool for time-series data, especially for time-series prediction or sequence to sequence learning. Convolutional neural networks (CNNs) on the other hand are mostly used for classification tasks on images. While RNNs do not apply very well to image data, there is no reason why CNNs would not apply in one dimension. If CNNs perform well in a classification task, like the one we present in our paper, they have one major advantage: They are much faster in training and inference (which was approximately a x40 speed-up in a preliminary test using a typical LSTM architecture) and also more stable during training. For real-time data processing with a short temporal resolution, how it is envisaged for CMLs, this is very important. Therefore, we want to emphasize that our results show that CNNs are a valid processing tool for one-dimensional data. On top of that, we believe that computational resources should be saved, unless absolutely necessary.

**4. ANN for rain rate estimation**

This is indeed an interesting topic. In fact, we already did such an evaluation and for now arrived with the following conclusions: The results are good, but not over-whelming which is probably due to the fact, that in the hourly RADOLAN reference the uncertainty of the absolute rainfall amount is much higher than the uncertainty of the temporal localization of an event. Learning absolute rain rates from this reference is therefore not our goal, since we want to use the attenuation-rain-rate (k-R) power law, which is very insensitive to DSD variations. This way, we avoid making absolute CML rainfall amounts mostly a "radar-adjusted" precipitation product. Another reason is that the wet antenna attenuation is still a big unknown, which should be investigated in the future and we do not want to mix this processing step with the detection or

the k-R relation, since there is an unknown risk of being "right for the wrong reason" when comparing to the weather radar. With the optimized event localization through the CNN and the near linear relationship between attenuation and rain rate, there is a promising chance of new insights into the wet antenna effect.

**5. Training and/or validation for different climates**

We appreciate this suggestion and we will do experiments for using CML subsets from different regions of Germany. Since CML data might be compromised during winter time, due to wet snow and ice covers on the antennas, we are, however, not sure if we can separate training and validation between winter and summer to simulate a large climatological difference. Regarding CML data from other climatic regions, we unfortunately do not yet have enough CML data for Burkina Faso and in general we are lacking reference data there. We just started with a project (AgRAIN) to improve both issues, but it will take some time to have an effect.

**References**

[1] Kim, M.-S. and B.H. Kwon, 2018: *Rainfall Detection and Rainfall Rate Estimation Using Microwave Attenuation*, Atmosphere, 9, 287, doi.org/10.3390/atmos9080287

[2] Habi, H. V. and H. Messer, 2018: *Wet-Dry Classification Using LSTM and Commercial Microwave Links*, IEEE 10th Sensor Array and Multichannel Signal Processing Workshop (SAM), doi.org/10.1109/SAM.2018.8448679

[3] de Vos, L.W., A. Overeem, H. Leijnse, and R. Uijlenhoet, 2019: *Rainfall Estimation Accuracy of a Nationwide Instantaneously Sampling Commercial Microwave Link Network: Error Dependency on Known Characteristics* J. Atmos. Oceanic Technol., 36, 1267–1283, doi.org/10.1175/JTECH-D-18-0197.1

---

## Author Response (AR1)

**Authors response to the Editor and the Referees for in regard of the revision of the paper:**

Dear Editor and Reviewers,

we appreciate your comments which we used as a basis to improve our manuscript. This response letter is structured in the following way. We first summarize our general changes which follow the recommendations of the reviewers. Then we describe further general changes that we made in the analysis. Additionally, we provide a copy of our final response to the reviewers (that we submitted to AMTD at the end of the open discussion) and a marked-up manuscript version of all changes.

The following changes have been introduced following the recommendations of the reviewers:

- The main issue raised during the discussion phase was a lack of scientific significance of the results we presented. To underline the importance of the processing step of rain event detection in CML signal levels we now additionally analyzed the impact of falsely detected and missed wet periods on the overall amount of rainfall estimated by CMLs. We show that the impact is large and that an improvement in the rain event detection directly influences the estimated rainfall amount. To our knowledge such an analysis has not been done before. To illustrate why fluctuations during dry periods constitute such a challenge for detection algorithms we added three exemplary time-series in Fig. 1. The introduction was revised in large parts focusing on a better review of challenges and state of the art. For better readability we separated the introduction into several sections. (Changes in revised version: revised section 1 and new sections 2.5, and 4.3 and an additional paragraph in section 3)

- We introduced different setups for the added 'old data' that we provide to the CNN as an input and which has the purpose of a reference to previous behavior of the CML. Previously we used 120 minutes and now added a larger analysis for 0, 15, 30, 45, 60, 120, 180 and 240 minutes. This way a ROC analysis in Fig. 4 c) can show that the performance increase converges with longer windows and significant changes are not visible beyond the 120 minute variant. As a consequence the main analysis keeps the simplest version with good performance, which is the 120 minute old data version. This addresses specific comment 2 of referee 1 and specific comment 2 of referee 2. (Changes in revised version: Added plot in Fig. 4, added Table A1 and new text in sections 2.3.1 and 3)

- We introduced an additional CNN architecture that uses the CML metadata as an input. The performance of this new model is also shown in Fig. 4 c). No increase in performance could be observed and the model was not used for further analysis. This addresses specific comment 3 of referee 1 and the comment of referee 2, that in principle, a model could make use of this information. (Changes in revised version: Added plot in Fig. 4 and new text in sections 2.3.1 and 3).

- Information about the dropout layers was added in the text as requested by referee 2. (L.230)

- The definition of TP, FP, FN and TN was clarified.  (Updated version of Table 1)

- In section 2.4. we added explanations about the different intended use cases for MCC and ROC as requested by referee 2. (L.283 and L.296)

- The remaining points have already been answered in our author's comment below.

- We corrected our error regarding the citation of Kim and Kwon (2018) .

As described in our final author response at the end of the open discussion, we did not consider all recommendations of the reviewers. These are the following:

- We did not include an investigation on LSTM networks in our analysis as we believe it is out of the scope of this paper to include another method in addition to the two that are discussed and evaluated here in detail. We already elaborated this topic in our direct answer to referee 2.

- We did not find an appropriate way of separating the data set into different climatic settings other than separating liquid and solid precipitation types. However, it is not our goal to investigate the transferability between those, because CMLs do not perform well when precipitation is solid or of mixed type. We will therefore wait to do this specific analysis (not only for rain event detection but CMLs in general) until we gain more data from other climatic regions from projects that are already starting now.

In addition to the changes that we introduced based on the reviewer suggestions, we made the following changes to our data processing:

- To increase the robustness of the normalization we increased the maximum time for the rolling median from 24 to 72 hours if that data is available.

- We increased the number of training CMLs from 400 to 800. Compared to the original results, there is not much of an improvement but this increased the number samples available when using the newly introduced 5 hour input data windows instead of 3.

- In accordance with the rain rate estimation scheme from Graf et al. (2019) we increased the time series interpolation from 3 to 5 minutes.

- We adjusted the initial learning rate, stopping rule and model selection resulting in a faster training time but reaching a similar performance.

Due to these changes, the absolute numbers of the results in this revised version differ from those in the initial version, but the conclusions remain the same. The largest change is due to the new threshold optimization using the unbalanced data set VALAPR, which makes more sense when comparing to the RSTD method which is also optimized for the original unbalanced data. This leads to a significant increase in the CNNs MCC which can be observed in Fig 7.

Due to the substantial overhaul of our manuscript, additional smaller changes are not listed here, but can be derived from the marked up manuscript version.

We believe that the substantial revision and the relevant additional analysis we conducted are now able to clearly demonstrate the scientific significance of rain event detection in CML data and the improvements our proposed CNN-based method provides.

**Final response from AC2**

This is our final response after the discussion phase and it addresses both referee comments. During the open discussion phase we already provided a quick, but comprehensive, response to referee 2 (Andreas Scheidegger). It was meant to provide feedback from our side to encourage further contributions. In the following, we will first give a summary of our assessment of the major issues, as they were reported by the two referees. We will then discuss these major issues and propose the related changes and additions for a revision of our manuscript. Finally we will list all individual comments of the referees and our corresponding responses, referring to our general answers where appropriate.

**Summary and assessment of major referee comments**

We thank both referees for their critical assessment of our manuscript. According to their comments they acknowledged that the manuscript is well written and that our analysis is scientifically correct. Both referees have pointed out major issues, though. According to the individual referee comments, these major issues are:

1. **Lack of scientific significance:** In contrast to referee 1, who rates our manuscript to have "good scientific significance", Andreas Scheidegger's main objection is "poor scientific significance" and he therefore recommends a rejection. While we clearly see that we have to improve on the justification of our research in the manuscript, we do not agree with this assessment. Our detailed response and the proposed changes for a revised manuscript can be found in the next section.

2. **Legitimacy of comparing to method based on rolling standard deviation:** The comparison with the rolling standard deviation (RSTD) method of Schleiss and Berne 2010 [1] is criticized by both referees, albeit for different reasons. Referee 1 states that an intercomparison with another neural network architecture, namely LSTM, is essential. Unfortunately, a clear reason for rejecting the comparison to the RSTD method is not visible from their argumentation. Andreas Scheidegger is concerned about the simplicity of our chosen reference method and that this diminishes scientific quality. Our main argument for using the method is that it is still state of the art. Thus, using it as a reference enables other CML researchers to put the performance gain into perspective. We explain this argumentation and our proposed changes for the manuscript below.

**Summarizing answers to major issues and proposed changes for a revision:**

**To 1.** Lack of scientific significance**:**
In our answer to Andreas Scheidegger's general comments on our work (See AC1 of the discussion) we already discussed the importance of rain event detection for the CML-rainfall community (backed up by supporting statements from other community members that deal with the same kind of CML data). In summary, we acknowledge that the relevance may be low from a machine learning perspective, but this is not a computer science paper and it was not submitted to a computer science journal. As explained in AC1 the relevance is given by the application, which is the improvement of quantitative precipitation estimation with commercial microwave links. To appear shallow in one discipline is a common hurdle for interdisciplinary research items, although interdisciplinary research is wanted by the scientific community. Our suggestions for improving our manuscript in order to highlight the relevance for the application are the following:

    i.    We will revise the introduction by adding a more detailed review of previously proposed methods for rain event detection, separating them into application on the different data acquisition types of min/max and instantaneous sampling. Methods developed for one kind of data acquisition are in general incompatible with the other kind. Whenever this information is

publicly available, we will review the problems previous methods had with false detections and missed events. While many previous studies were event based evaluations, our study is free of any preselection which is a necessary analysis for potential operational applications.

ii. To show the significant impact of rain event detection on estimated rain rates, we will add an analysis about the rainfall overestimation through falsely classified rain events and underestimation through missed events in absolute numbers. Using the same processing scheme as in Graf et al. 2019 we will discuss the improvement over the previous method in terms of absolute hourly and monthly rainfall amount. A preliminary evaluation is shown in the plot below from which we can derive a reduction of the overestimation through falsely classified rain events by 27% of the monthly overestimation through the Q80 method. Additionally the CNN reduces the underestimation through missed events by 15%. A larger evaluation and description of the derivation of the final rain rates will be added to the revised manuscript.

[Figure]

Figure 1: Histogram of false positive hours (FP) in April 2018 (reference rain rate below 0.1 mm). At a threshold of 0.8 the CNN reduces the number of falsely detected rain events and therefore the total falsely generated rainfall amount by approximately 27%. At the same time the CNN still misses less events than the standard deviation method (not shown in this plot).

**To 2.** The comparison to the method of Schleiss and Berne 2010 [1].

In our opinion, the comparison is justified as follows:

First, the method should still be considered state of the art due to the works of Fencl et al. 2020 [3], Graf et al. 2019 [4], and Kim and Kwon 2018 [5] using the rolling standard deviation according to [1] for rain event detection. Additionally De Vos et al. 2019 [6] are using correlation to nearby links and Ostrometzky and Messer 2018 [7] propose a simple rolling minimum to set the baseline level. Both methods are of similar or lower complexity than the RSTD method which is still applied to CML attenuation data due to its robustness (in the sense that only few parameters have to be tuned) and easy applicability. In our work, we show a significant improvement over the RSTD, while the resulting model is just as easy to apply using our pre-trained model, which we also share as open source software.

Second, although very simple, the RSTD method is not performing poorly. In a Lab setup rain events could be purely detected using the rolling standard deviation due to the justified assumption that fluctuations are bounded during dry periods and they exceed the boundary even for small rain events.

Unfortunately, this does not always apply to real world data, where strong signal fluctuations occur due to e.g. multipath effects. The challenge is to separate those artifacts from real rainfall fluctuations, and the fact that Andreas Scheidegger is not surprised that the CNN can outperform the RSTD solidifies our assumption that we picked the right model for the large scale real world application.

We believe that this justifies our choice of comparing to the RSTD method from [1]. We propose to revise our introduction by citing the works that underline the state of the art status of using the RSTD method and we will add more examples of real world CML attenuation time-series that underline the challenges which can not be solved by this method, thus generating the knowledge (or performance) gap, that our method seeks to fill.

The question which artificial neural network architecture should be used is therefore of secondary relevance. Previously, LSTM was used for min/max sampled attenuation data and a small amount of CMLs [2]. The main point of our study is showing the potential of an artificial neural network approach for instantaneously sampled data from a large amount of sensors likely to occur in an operational setup and certify the robustness for further application. This was not done before. Again, we believe that showing this potential by comparing to the state of the art is justified, if not necessary.

**Direct answers to referee comments:**

Note: The complete referee comments are copied here using italic font, our response uses normal font.

**Anonymous Referee #1**

*In general, this paper demonstrates the use of a 1D convolution neural network for the task of Wet-Dry classification using commercial microwave links (CMLs). The scientific significance of this paper is in presenting the potential of the suggested method for the specific application: the use of 1D CNN for wet-dry classification with commercial microwave links. But, without any theoretical justification for the use of 1D CNN, it must be compared empirically with other algorithms/methods. The results are shown in this work only compare 1D CNN with a model-driven method [1], however, the suggested method must be compared with another data-driven algorithm, previously suggested (and cited by the author) - the use of LSTM for wet-dry classification [2]. This comparison is important also since the LSTM can capture long sequence, while the CNN only see a fix window size of the attenuation time series. Additionally, the authors didn't use the CML's parameters (e.g. link length, frequency, and polarization ) as an additional input to the neural network, which may make this method more sensitive to differences in those parameters.*

We thank the referee for their critical assessment of the comparison to the RSTD method which is used to show the potential of our proposed method. Our justification for doing this comparisonis given in paragraph 2 above. The use of LSTM is discussed in 1

*Specific comments required for the paper to be acceptable for publication:*

*1. Comparing the results of the LSTM and CNN on the same data set is essential.*

To 1.: While it is true that LSTM is a common neural network architecture which might also be applicable to the task we miss a fact based justification for 'must be compared with another data-driven algorithm'. As written above, the use of a rolling standard deviation can be considered state of the art. The method makes heavy use of long term CML statistics and can be called a statistical method. We therefore do not understand what the word 'model-based' should mean in this context or why the comparison to the RSTD method should be unfair or of less scientific relevance. Apart from that, our justification for using CNNs is the generally accepted fact that they are good in recognizing patterns independent of their location within a longer sequence of the time-series and this is also what we wrote in the manuscript (see lines 8, 67 and 158). As we already stated in paragraph 3 of AC1, LSTM is a common ANN architecture for time series analysis. This does not mean that CNNs, which have the benefit of very fast parallel processing, are not applicable to time-series data. Indeed, our results prove that CNNs are a valid processing tool for one-dimensional data.

*2. Study the effect of different window sizes on the performance of the proposed method.*

We evaluated the effect of longer window sizes and did not find a significant improvement. But we agree that information about suitable window sizes is important. Hence, in the revised version, we will shortly describe the different setup with a 5 hour window length and add the results to table 2.

*3. Study the effect of different CML's parameters (e.g. link length, frequency, and polarization).*

We evaluated the effect of adding the CML parameters and did not find a significant improvement. According to the k-R power-law, CML frequency and length influence the amplitude of the rainfall-induced fluctuations. E.g. short CMLs with a comparably low frequency are less sensitive to rainfall. We expected that this information about the CML sensitivity to rainfall would help the classification performance. We did not see a relevant improvement, though. One possible explanation is that knowing the CML parameters does not help to detect wet periods close to or below the individual CML's detection limit, since there just might not be a detectable signal. Furthermore, according to our experience with CML time series, the occurrence of signal fluctuations during dry periods, for which distinguishing between wet and dry is most challenging, does not depend on CML parameters.

Initially, we decided to only describe the less complex and equally well performing setup without the CML parameters for the sake of brevity.

We will now include our results from the CNN with the CML parameters in the revised manuscript, by shortly describing the different setup and adding the results to table 2.

**Andreas Scheidegger:**

*\* General*
*The manuscript describes the application of a one-dimensional convolutional neuronal network (CNN) to classify wet and dry periods based on microwave link attenuation data. The CNN is compared against a very simple classification scheme that is only based on the standard deviation of the signal. Not surprisingly, the CNN performed better.*

*The manuscript is well written and the underlying work seems solid. Still, in my opinion, this paper lacks ambition and innovation to deserve a publication in AMT. As the authors mention, various ANN's and other machine learning techniques have previously been applied in different settings on MWL data. Also the whet/dry classification problem does not appear particulary challenging from a machine learning perspective. Furthermore, for time-series data recurrent neuronal network architectures (e.g.LSTM) seem a more obvious choice (which could be combined with convolution layers if needed).*

This is, in our opinion, the main issue which we discuss in our summarizing answer 1. above and in the majority of AC1. In summary, rain event detection is a necessary processing step to set the baseline signal level, required to derive the specific attenuation $k$ which is then used to derive the rain rate $R$ via the k-R relation. The community state of the art are methods like the RSTD and the problems with false detections can be shown from recent publications. Although less relevant from a machine learning point of view, the relevance for the application is high.

*A more interesting question would be to investigate if we can train a ANN to predict the rainfall intensities directly, and so avoid all submodels for baseline estimation, wetantenna correction, and so on. Such a model could also make use of additional information, like MWL properties, frequency, maybe temperature, …*

This is answered in AC1 paragraph 4.

*A good transferability of the trained model to a region with different climate is key for an application where no reference data are available (such as in the mentioned Burkina Faso). This could have been partly emulated by training the model in one region and then validating it in a region with different climate. Or by training the model in winter and validating it in summer.*

This is answered in AC1 paragraph 5.

*I'm sure the current work offers the authors a solid foundation for more ambitious investigations.*

No answer required.

*L 90: "...it has to be proven that artificial neural networks allow for high-performance,fast and robust processing of large data sets..." - I think this is already proven by countless other application.*

This statement has to be read in the context of CML data sets. The purpose is not to show that CNNs can achieve this in general, which is indeed proven on countless examples. To show that the behaviour of a large CML data set can be predicted by training only on a comparably small subset is still to be proven for the application and the CNN is our method of choice to do so. To our knowledge no previous study achieved or even investigated this degree of generalization. In fact, most previous studies use a setup that uses long time statistics of all individual CMLs for a low amount of CMLs and

many of them deal with min/max sampled data, which is different from our instantaneous measurements. We will adjust the relevant paragraph in the manuscript accordingly, to better explain that.

*L 145: Besides the attenuation data for the hour to classify, the Network was also feed with the two hours of "old" data. Did this improve the classification? If yes, it would indicate that attenuation data have some kin of memory effect (antenna wetting?).*

Adding the two hours of "old" data has a positive effect. Antenna wetting is kind of a memory effect since it will increase the baseline level during the rain events, but it will also keep the baseline level increased after the rain event. With the drying of the antenna, the baseline level then decreases slowly after the event. Since this drying process is quite continuous it does not lead to strong rain-like attenuation fluctuations of the signal level. Hence, we think, the wet antenna effect is not the reason for the improvement with the added 'old' data. Our reasoning for using 'old' data was that this data provides more context for the CNN, i.e. there is a lot more information on how a CML time series generally 'behaves'. Also as a human it is a lot easier to distinguish rain events from dry fluctuations when the available time period is larger. Humans are, compared to CNNs, unfortunately very slow in doing so. In addition to the longer window size, as requested by reviewer 1, we will include the performance when not providing any old data to show this improvement.

*L185: Where did you add the dropout layers? How many?*

Dropout was used between the fully connected layers, i.e. two times. We will indicate the exact location in figure 2.

*L216: Are TP, FP, FN, and TN defined?*

We introduced this notation in table 1. We will add a direct reference to the table at the first occurence of TP, FP, FN, and TN. We will also add a set theoretic definition in addition to the table.

*L230: What is the advantage of the MCC compared to the ROC?*

The Matthews Correlation Coefficient is a single number, which can be used to optimize the threshold for the CNN or the thresholds of the rolling standard deviation. The ROC consists of the two numbers FPR and TPR. Using them for optimizing a threshold is not straight forward and would require a cost function that can be minimized. The advantage of ROC is that the performance of classifiers with a variable threshold can be compared independent of a fixed threshold value by considering the ROC curve. We will clarify these different purposes in the respective parts of the method section.

[revised manuscript text omitted]